# Trivialized Momentum Facilitates Diffusion Generative Modeling on Lie Groups

**Yuchen Zhu**[*], **Tianrong Chen**[*], **Lingkai Kong, Evangelos A. Theodorou, Molei Tao**[†]
Georgia Institute of Technology
`{yzhu738,tchen429,lkong75,evangelos.theodorou,mtao}@gatech.edu`

## Abstract

The generative modeling of data on manifolds is an important task, for which diffusion models in flat spaces typically need nontrivial adaptations. This article demonstrates how a technique called 'trivialization' can transfer the effectiveness of diffusion models in Euclidean spaces to Lie groups. In particular, an auxiliary momentum variable was algorithmically introduced to help transport the position variable between data distribution and a fixed, easy-to-sample distribution. Normally, this would incur further difficulty for manifold data because momentum lives in a space that changes with the position. However, our trivialization technique creates a new momentum variable that stays in a simple fixed vector space. This design, together with a manifold preserving integrator, simplifies implementation and avoids inaccuracies created by approximations such as projections to tangent space and manifold, which were typically used in prior work, hence facilitating generation with high-fidelity and efficiency. The resulting method achieves state-of-the-art performance on protein and RNA torsion angle generation and sophisticated torus datasets. We also, arguably for the first time, tackle the generation of data on high-dimensional Special Orthogonal and Unitary groups, the latter essential for quantum problems. Code is available at `https://github.com/yuchen-zhu-zyc/TDM`.

## 1 Introduction

Diffusion-based (e.g., Song et al., 2020b; Ho et al., 2020; Dhariwal & Nichol, 2021) and flow-based (e.g., Lipman et al., 2022; Liu et al., 2023; Albergo & Vanden-Eijnden, 2023) generative models have significantly impacted the landscape of various fields such as computer vision, largely due to their remarkable ability in modeling data that follow complicated and/or high-dimensional probability distributions. However, in many application domains, data explicitly reside on manifolds. Note this is different from the popular data manifold assumption which is implicit; here the manifold is a priori fixed due to, e.g., physics. Such cases occur, for example, in protein modeling (Shapovalov & Dunbrack, 2011; Yim et al., 2023b; 2024; Bose et al., 2024), cell development (Klimovskaia et al., 2020), geographical sciences (Thornton et al., 2022), robotics (Sola et al., 2018), and high-energy physics (Weinberg, 1995). The naive application of standard generative models to these cases via embedding data in ambient Euclidean spaces often results in suboptimal performance (De Bortoli et al., 2022). This is partly due to the lack of appropriate geometric inductive biases and potential encounters with singularities (Brehmer & Cranmer, 2020).

Pioneering works suggest generalizing (continuous) neural ODE (Chen et al., 2018) to manifolds (Mathieu & Nickel, 2020; Lou et al., 2020; Falorsi & Forré, 2020) with maximum-likelihood training. Rozen et al. (2021); Ben-Hamu et al. (2022) develop simulation-free algorithms but their objective is unscalable or biased (Lou et al., 2023). Recent milestones, such as Riemannian Score-based Model (RSGM) De Bortoli et al. (2022), Riemannian Diffusion Model (RDM) (Huang et al., 2022), and Riemannian Diffusion Mixture (Jo & Hwang, 2023) have successfully demonstrated the potential to extend diffusion models onto Riemannian manifolds. RSGM explores the effectiveness and complexity of various variants of score matching loss on a general manifold and their applicable

---

[*]Equal contribution.
[†]Corresponding author.

Figure 1: Visualization of algorithmic intuition of TDM. Existing approaches such as RFM and RSGM often model an object that lies on changing tangent spaces as the position $g_t$ moves, resulting in inaccuracies when handling complicated manifolds during trajectory simulations. In contrast, TDM only needs to learn the score in **simple Euclidean space**. Thanks to the special structure of trivialization, TDM guarantees the induced momentum will strictly lie on the tangent space, which improves generation quality and reduces sampling error.

scenarios, and RDM discusses techniques such as variance reduction for the training objective via importance sampling and likelihood estimation. Building upon RSGM and RDM, Riemannian Diffusion Mixture further leverages a mixture of bridge processes to achieve a significant improvement in training efficiency. These models learn to reverse the diffusion process on a manifold. This is achieved through the employment of Riemannian score-matching methods, which serve as simulation-based objectives for the optimization of the model. However, due to the inherent geometric complexity of the data, the training and sampling processes of such models necessitate multiple approximations. In particular, they require the projection of the vector field (i.e. score) to the tangent space which subsequently serves as the training label for the neural network during the training phase. Furthermore, to mitigate numerical integration errors during the sampling process, there is a requirement for the projection of samples to the original data manifold. Moreover, among most scenarios, as these models are simulation-based algorithms, an additional approximation is introduced during the collection of training data from the simulation during training. This process also necessitates the projection of data to the manifold, which is analogous to the sampling phase. The combination of all these three approximations can compromise the quality of generation.

Even more recent advancements, such as Riemannian Flow Matching (RFM) (Chen & Lipman, 2024) and Scaled-RSGM (Lou et al., 2023), aim to alleviate training complexities and enhance model scalability through the introduction of simulation-free objectives. Scaled-RSGM achieves this by focusing on Riemannian symmetric spaces, while RFM constructs conditional flows using premetrics. However, it is important to note that both of the approaches still require some of the aforementioned approximations during the training and sampling phases (such as projections onto tangent spaces), which may potentially introduce inaccuracies in the generated results. Furthermore, whether RFM is expressive enough for intricate data distribution was discussed in (Lou et al., 2023).

In this work, we build upon recent progress in momentum-based optimization (Tao & Ohsawa, 2020) and sampling (Kong & Tao, 2024) on Lie groups to develop a highly scalable and effective generative model for data on these manifolds, named **T**rivialized **D**iffusion **M**odel (TDM). Our approach departs from prior momentum-based generative models (Dockhorn et al., 2021; Chen et al., 2023; Pandey & Mandt, 2023; Chen et al., 2024) due to an additional technique called trivialization, which utilizes the additional group structure and enables us to learn score in a fixed flat space, while still encapsulating the curved geometry without any approximation.

It's also worth noting that several works have already achieved success in the generative modeling of data distribution on special Lie groups such as $\mathsf{SE}(3)$ and $\mathsf{SO}(3)$, resulting in a remarkable performance on applications like protein backbone generation (e.g., Yim et al., 2023b;a; Bose et al., 2024). The seminal work of FrameDiff (Yim et al., 2023b) extends RSGM to $\mathsf{SE}(3)$ and $\mathsf{SO}(3)$ and takes advantage of the pleasant properties of heat kernel $\mathsf{IGSO}(3)$ for $\mathsf{SO}(3)$ to perform denoising score matching. Due to the tractable computation of the heat kernel, FrameDiff also enjoys the benefits of efficient learning in a fixed space as well as projection-free simulation on the manifold, much similar to the advantages enjoyed by TDM. However, we need to clarify that the success of FrameDiff, while sharing a similar spirit with TDM, comes from distinct sources. FrameDiff leverages the special structures of $\mathsf{SO}(3)$ and its algorithm can't be easily generalized to other cases. In contrast, TDM holds the advantage of efficient learning and projection-free sampling for general Lie groups (see, e.g., Sec.3 for quantum applications of other Lie groups).

To sum up, our contributions can be summarized into the following three bullet points:

1) We introduce TDM that enables manifold data generative modeling through learning a trivialized score function in a fixed flat space, which dramatically improves the generative performance.

2) We leverage a nontrivial Operator Splitting Integrator to stay exactly on the manifold in an accurate and efficient way. The reduction of approximations further improves the generation.

3) We outperform baselines by a large margin on protein/RNA torsion angle datasets. We achieve much higher quality generation on a newly-introduced challenging problem called Pacman. We present the first results on generating $\mathsf{U}(n)$ data corresponding to quantum evolutions, and high dim $\mathsf{SO}(n)$ data too; these results are also appealing.

## 2 METHOD

In this section, we will discuss how to perform generative modeling of data distribution on a class of smooth Riemannian manifolds, namely Lie groups, by learning a score function similar to a Euclidean one. Our goal is to recover the scenario of Euclidean generative modeling to the maximum by leveraging the group structure of the Lie group apart from its Riemannian manifold structure. To achieve this, we explore a specific manifold extension of Kinetic Langevin dynamics Nelson (1967), which contains an additional variable known as momentum. Importantly, a direct introduction of the momentum would not simplify the situation, since the momentum lives in a changing tangent space as the position moves. Fortunately, the group structure of the Lie group enables the design of a trivialized momentum that stays in a Lie algebra for the whole time, which is a **simple fixed Euclidean space** that suits our needs. In the following, we will discuss how the technique of trivialization can help completely avoid challenges posed by the curved geometry in an exact, analytical fashion, without resorting to complicated differential geometry notions such as parallel transport, and certainly no need for approximations, projections, and retractions.

In the following, we first introduce a forward process that converges to an easy-to-sample distribution with such trivialized momentum. We derive the time reversal of such a process, which can serve as a backward generative process. We discuss methods to efficiently learn the drift of the backward process. Finally, we introduce a numerical integrator that achieves high accuracy and preserves the manifold structure of the Lie group.

We also provide a brief review of the Euclidean Diffusion Models, kinetic Langevin dynamics, and Lie group. For details, please see Appendix A for more information.

---

**Algorithm 1** TDM (Trivialized Diffusion Model)

---

**Require:** Iteration $N_{\text{iter}}$, Total time horizon $T$, Simulation steps $N$, time step $h = T/N$, parameter initialization $\theta_0$, Lie group data $\{g^m\}_{m=1}^M$, friction constant $\gamma > 0$, early-stopping time $\varepsilon$
  **// TRAINING**
 1: **for** $n = 0, \ldots, N_{\text{iter}} - 1$ **do**
 2:    Sample $\bar{g} \sim \frac{1}{M} \sum_{m=1}^M \delta_{g^m}$         ▷ Sample initial $g$ from data
 3:    Sample $\bar{\xi}$ by i.i.d. generate $\bar{\xi}^i \sim \mathcal{N}(0,1)$ for $1 \le i \le \dim \mathfrak{g}$   ▷ Sample arbitrary initial $\xi$
 4:    **if** $\mathcal{J}_{\text{DSM}}$ is tractable **then**           ▷ Use DSM if possible
 5:      Sample $t \sim \text{Uniform}[\varepsilon, T]$, $g_t, \xi_t \sim p_{t|0}(g, \xi | \bar{g}, \bar{\xi})$
 6:      $\ell(\theta_n) = \mathcal{J}_{\text{DSM}}(\theta_n, \{g_t, \xi_t\})$    ▷ Compute Denoising score matching objective
 7:    **else**                    ▷ Use ISM instead
 8:      $\{g_t, \xi_t\} = \textbf{FSOI}(\bar{g}, \bar{\xi}, \gamma, h, N)$    ▷ Simuate forward dynamic with Algorithm 2
 9:      $\ell(\theta_n) = \mathcal{J}_{\text{ISM}}(\theta_n, \{g_t, \xi_t\})$     ▷ Compute Implicit score matching objective
 10:    **end if**
 11:    $\theta_{n+1} = \text{optimizer\_update}(\theta_n, \ell(\theta_n))$       ▷ AdamW optimizer step
 12: **end for**
 13: Set optimal $\theta^* = \theta_{N_{\text{iter}}}$
  **// SAMPLING**
 14: Sample $(g_0, \xi_0) \sim \pi_*$     ▷ Sample initial condition from stationary measure
 15: $(g_N, \xi_N) = \textbf{BSOI}(g_0, \xi_0, s_{\theta^*}, \gamma, h, N)$    ▷ Simulate backward dynamic with Algorithm 3
 16: **return** $\theta^*, (g_N, \xi_N)$

---

## 2.1 TRIVIALIZED KINETIC LANGEVIN DYNAMICS ON LIE GROUP AS NOISING PROCESS

A Lie group is a manifold with a group structure, which gives us an important tool called left-trivialization to handle momentum. The left group multiplication $\mathsf{L}_g : G \to G$ is defined as $\mathsf{L}_g : \hat{g} \to g\hat{g}$, whose tangent map (also known as differential) $T_{\hat{g}}\mathsf{L}_g : T_{\hat{g}}G \to T_{g\hat{g}}G$ is a one-to-one map. As a result, for any $g \in G$, we can represent the vectors in $T_g G$ by $T_e\mathsf{L}_g\xi$ for any $\xi \in T_e G$, where $e$ is the group identity and $\mathfrak{g} := T_e G$ is the Lie algebra.

Utilizing such property, Kong & Tao (2024) appropriately added noise to variational Lie group optimization dynamics (Tao & Ohsawa, 2020) and constructed the following kinetic Langevin sampling dynamics on Lie groups:

$$\begin{cases} \dot{g}_t = T_e\mathsf{L}_{g_t}\xi_t, \\ \mathrm{d}\xi_t = -\gamma(t)\xi_t\mathrm{d}t - T_g\mathsf{L}_{g_t^{-1}}(\nabla U(g_t))\mathrm{d}t + \sqrt{2\gamma(t)}\mathrm{d}W_t^{\mathfrak{g}}, \end{cases} \tag{1}$$

where $(g_t, \xi_t) \in G \times \mathfrak{g}, \ \forall t \geq 0$, here $G$ denotes a Lie group and $\mathfrak{g}$ denotes its associated Lie algebra, $\mathrm{d}W_t^{\mathfrak{g}}$ is the Brownian motion on Lie algebra $\mathfrak{g}$, $\nabla U$ is the Riemannian gradient of $U$, and $U : G \to \mathbb{R}$ is a potential function. $\xi_t$ is the left-trivialized momentum at time $t$ and $T_e\mathsf{L}_{g_t}\xi_t$ is the true momentum.

They also proved (Kong & Tao, 2024) that for connected compact Lie groups, which will be our setup, (1) converges, under Lipschitzness of $\nabla U$, exponentially fast to its invariant distribution, which is

$$\pi_*(g, \xi) = \frac{1}{Z}\exp\left(-U(g) - \frac{1}{2}\langle\xi, \xi\rangle\right)\mathrm{d}g\mathrm{d}\xi, \tag{2}$$

where $\mathrm{d}g$ denotes the Haar measure, $\mathrm{d}\xi$ denotes the Lebesgue measure on $\mathfrak{g}$, and $Z$ is the normalizing constant. Dynamic (1) is a generalization of the Euclidean kinetic Langevin equation on $\mathbb{R}^k$ to general Lie groups ($\mathbb{R}^k$ is a Lie group with vector addition being the group operation).

By Peter-Weyl Theorem, a connected compact Lie group can be represented as a closed subgroup of $\mathrm{GL}(n, \mathbb{C})$ (Knapp, 2002), i.e. the group of $n \times n$ invertible matrices with entries in $\mathbb{C}$. Such representation can be computed explicitly by, e.g., adjoint representation (Hall, 2013). When $g_t$ and $\xi_t$ are both represented as matrices, the abstractly defined $T_e\mathsf{L}_{g_t}\xi_t$ in (1) can be calculated explicitly by a matrix multiplication between the matrix representation of $g_t$ and $\xi_t$, where we write it as $g_t\xi_t$. In the rest of this work, to avoid confusion and simplify notations, we use $g_t$ and $\xi_t$ to refer to their corresponding matrix representation and use $g_t\xi_t$ to denote $T_e\mathsf{L}_{g_t}\xi_t$.

We want to construct a forward noising process based on (1) by choosing a potential $U$ that corresponds to an easy-to-sample distribution. In the case of connected compact Lie groups, we pick the natural choice, which is $U(g) = 0, \forall g$, corresponding to an invariant distribution whose $g$ marginal is the uniform distribution on $G$, or more precisely, the Haar measure. In this case, the following dynamics would be the forward noising process,

$$\begin{cases} \dot{g}_t = g_t\xi_t, \\ \mathrm{d}\xi_t = -\gamma(t)\xi_t\mathrm{d}t + \sqrt{2\gamma(t)}\mathrm{d}W_t^{\mathfrak{g}}, \end{cases} \tag{3}$$

Important examples of connected compact Lie Groups include but are not limited to the Special Orthogonal group $\mathsf{SO}(n)$, the Unitary group $\mathsf{U}(n)$, the Special Unitary group $\mathrm{SU}(n)$, etc. Note 1-sphere $\mathbb{S}^1$, torus $\mathbb{T}$, and $\mathsf{SO}(2)$ are essentially the same thing (isomorphic). Note also the direct product of any two connected compact Lie groups is still a connected compact Lie group, so in general we can consider the Lie group $G$ of form, for $G_1, \ldots, G_k$ connected compact Lie groups

$$G = G_1 \times G_2 \times \cdots \times G_k, \tag{4}$$

## 2.2 TIME REVERSAL OF TRIVIALIZED KINETIC LANGEVIN

The following result allows us to revert the time of the forward noising process. Thanks to the introduction of momentum and the fact that it is trivialized, the time reversal will be very similar to the Euclidean version Dockhorn et al. (2021) despite that $g$ lives on a manifold. This pleasant feature is because the forward dynamics (1) has no (direct) noise on $g$ dynamics and therefore no score-based correction is needed for its reversal, and $\xi$ on the other hand is simply a Euclidean variable. More precisely, one important implication of the momentum trivialization is that the only score present

in the dynamic is $\nabla_\xi \log p_{T-t}(g_t, \xi_t)$, which now stays in the Lie algebra $\mathfrak{g}$ (a **fixed space** and also is isomorphic to Euclidean space). This implies that we manage to get rid of $\nabla_g \log p_{T-t}$ from the dynamic, which is a much more complicated subject than $\nabla_\xi \log p_{T-t}$ due to being a Riemannian gradient and has complicated geometric dependency. This trivialization technique leads to benefits on numerical accuracy and score representation learning, which is not enjoyed by previous works such as RFM (Chen & Lipman, 2024), RDM (Huang et al., 2022) and RSGM (De Bortoli et al., 2022). More details of the advantages of the trivialized dynamic will be provided in Section 2.4.

---

**Theorem 1** (**Time Reversal of Trivialized Kinetic Langevin on Lie Group**). *Let $T \geq 0$, $W_t^{\mathfrak{g}}$ be a Brownian motion on the Lie algebra $\mathfrak{g}$. Let $\mathbf{X_t} = (g_t, \xi_t)$ be the trajectory of the forward dynamics* (3), *with $\mathbf{X}_t$ admitting a smooth density $p_t(g_t, \xi_t)$ with respect to the Haar measure on $G$ and Lebesgue measure on $\mathfrak{g}$. Then, the solution to the following SDE*

$$\begin{cases} \dot{g}_t = -g_t \xi_t, \\ \mathrm{d}\xi_t = \gamma(T-t)\xi_t \mathrm{d}t + 2\gamma(T-t)\nabla_\xi \log p_{T-t}(g_t, \xi_t)\mathrm{d}t + \sqrt{2\gamma(T-t)}\mathrm{d}W_t^{\mathfrak{g}}. \end{cases} \quad (5)$$

*satisfy $\mathbf{Y}_t \overset{d}{=} (\mathbf{X}_{T-t})$ under the notation $\mathbf{Y}_t := (g_t, \xi_t)$ and initialization $\mathbf{Y}_0 = \mathbf{X}_T$.*

---

Note although a similar time reversal formula has been given for the Euclidean case in (Dockhorn et al., 2021), their results are not applicable due to the presence of the manifold structure. In fact, we need a non-trivial adaptation of the arguments and the proof relies on the Fokker-Planck equation on the manifold $G \times \mathfrak{g}$. For details of the proof of Theorem 1, see Appendix B.

In addition, similar to standard Euclidean diffusion model, dynamic 5 also has a corresponding probabilistic ODE counterpart, given by the following:

---

**Remark 1** (Probability Flow ODE). *The following dynamic has the same marginal as* (5)

$$\begin{cases} \dot{g}_t = -g_t \xi_t, \\ \mathrm{d}\xi_t = \gamma(T-t)\xi_t \mathrm{d}t + \gamma(T-t)\nabla_\xi \log p_{T-t}(g_t, \xi_t)\mathrm{d}t \end{cases} \quad (6)$$

*as long as initial conditions are consistent.*

---

## 2.3 Likelihood Training and Score-Matching for Trivialized Kinetic Langevin

To perform generative modeling of data distribution, we would like to simulate and sample from the stochastic dynamic in (5). However, the score $\nabla_\xi \log p_{T-t}(g_t, \xi_t)$ is intractable and we want to approximate it with a neural network score model $s_\theta(g_t, \xi_t, t)$. We denote the sequence of probability distribution $q_t^\theta$ as the density of $\mathcal{L}(\mathbf{Y}_t^\theta)$ with respect to the reference measure, where $\mathbf{Y}_t^\theta$ is the trajectory of the following dynamic,

$$\begin{cases} \dot{g}_t = -g_t \xi_t, \\ \mathrm{d}\xi_t = \gamma(T-t)\xi_t \mathrm{d}t + 2\gamma(T-t)s_\theta(g_t, \xi_t, t)\mathrm{d}t + \sqrt{2\gamma(T-t)}\mathrm{d}W_t^{\mathfrak{g}}, \end{cases} \quad g_0, \xi_0 \sim \pi_*. \quad (7)$$

In order to generate new data with dynamic (7), we need $q_T^\theta \approx p_0$, which would require learning a score that is close to the true score $\nabla_\xi \log p_t(g, \xi)$. A natural starting point for learning the score is through Score Matching (SM) between $s_\theta$ and $\nabla_\xi \log p_t(g, \xi)$, but that alone is intractable because $p_t$ is not known a priori. Instead, we can directly extrapolate some classical tractable variants of SM, such as Denoising Score Matching (**DSM**) or Implicit Score Matching (**ISM**), to the Lie group case.

**Denoising Score Matching**    Note the score matching objective can be rewritten as (Vincent, 2011)

$$\mathcal{J}_{\mathrm{SM}}(\theta) = \underbrace{\mathbb{E}_{t,p_0}\mathbb{E}_{p_{t|0}}\left[\left\|\nabla_\xi \log p_{t|0}(g, \xi) - s_\theta(g, \xi, T-t)\right\|^2\right]}_{\mathcal{J}_{\mathrm{DSM}}(\theta)} + C_1$$

where $C_1$ is a constant independent of $\theta$. Hence, $\operatorname{argmin}_\theta \mathcal{J}_{\text{SM}} = \operatorname{argmin}_\theta \mathcal{J}_{\text{DSM}}$, but evaluating $\mathcal{J}_{\text{DSM}}$ only requires knowledge of the conditional transition probability $p_{t|0}$. The question boils down to finding out such condition transition probability induced by the forward dynamic (3).

Note that the Lie algebra $\mathfrak{g}$ is a tangent space of $G$ at the identity, so it's a vector space that is isomorphic to Euclidean space $\mathbb{R}^d$, where $d = \dim(\mathfrak{g})$. For example, the $\mathfrak{so}(2)$ is the Lie algebra of the Special Orthogonal group $\mathsf{SO}(2)$. $\mathfrak{so}(2)$ consists of all the $2 \times 2$ skew-symmetric matrices. This implies that, for any $\xi \in \mathfrak{so}(2)$,

$$\xi = \begin{bmatrix} 0 & \theta \\ -\theta & 0 \end{bmatrix}, \theta \in \mathbb{R} \implies \mathfrak{so}(2) \cong \mathbb{R}$$

Here, the Brownian motion $\mathrm{d}W_t^{\mathfrak{g}}$ on $\mathfrak{g}$ should be understood as $\mathrm{d}W_t^{\mathfrak{g}} = \sum_{i=1}^d \mathrm{d}W_t^i \cdot e_i$, where $\{\mathrm{d}W_t^i\}_{i=1,\dots,d}$ are independent standard Brownian motions on $\mathbb{R}$ and $\{e_i\}_{i=1,\dots,d}$ is an orthogonal basis for $\mathfrak{g}$. Therefore, the forward dynamic (3) with initial condition $g(0) = g_0, \xi(0) = \xi_0$ is equivalent to the following,

$$\begin{cases} \dot{g}_t = g_t \xi_t, \\ \mathrm{d}\xi_t^i = -\gamma \xi_t^i \mathrm{d}t + \sqrt{2\gamma}\mathrm{d}W_t^i \quad \forall 1 \leq i \leq d. \end{cases} \quad s.t \ g(0) = g_0, \ \xi^i(0) = \xi_0^i \quad \forall 1 \leq i \leq d. \quad (8)$$

Here, without loss of generality, we choose $\gamma(t)$ to be a constant $\gamma > 0$. We notice that each $\xi^i$ follows is OU process with an explicit solution. This reduces problem (8) to a matrix-valued initial value problem (IVP) for $g_t$, since $\xi_t$ can be treated as a known function of time. Then the IVP $\dot{g}_t = g_t \xi_t, g(0) = g_0$ is just a linear system.

Unfortunately, note that even though the linearity ensures linear structure in the solution, namely $g(t) = g_0 \Phi(t)$ where $\Phi$ is known as a fundamental matrix, $\Phi$ in general may not be analytically available in closed-form because the linear system has a time-dependent coefficient matrix. This differs from the scalar case where $\Phi(t)$ would just be $\exp(\int_0^t \xi(s)\mathrm{d}s)$ or the constant coefficient matrix case where $\Phi(t)$ would just be $\operatorname{expm}(\xi t)$. Instead, we can represent the solution using geometric tools, resulting in Magnus expansion Magnus (1954) in the following form

$$g(t) = g_0 \operatorname{expm}(\Omega(t)), \quad \Omega(t) = \sum_{k=1}^\infty \Omega_k(t). \quad (9)$$

Here $\{\Omega_k\}_{k=1,\dots,\infty}$ is called the Magnus series, which is written in terms of integrals of iterated Lie algebra between $\xi(t)$ at different times. The first three terms of the Magnus series are given below to illustrate the idea,

$$\Omega_1(t) = \int_0^t \xi(t_1)\mathrm{d}t_1, \quad \Omega_2(t) = \frac{1}{2}\int_0^t \int_0^{t_1} [\xi(t_1), \xi(t_2)]\mathrm{d}t_2\mathrm{d}t_1$$

$$\Omega_3(t) = \frac{1}{6}\int_0^t \int_0^{t_1} \int_0^{t_3} \left( [\xi(t_1), [\xi(t_2), \xi(t_3)]] + [\xi(t_3), [\xi(t_2), \xi(t_1)]] \right)\mathrm{d}t_3\mathrm{d}t_2\mathrm{d}t_1$$

In general, the solution given in (9) may not be tractable due to the fact that $\Omega(t)$ is an infinite series with increasing intricacy for each term. However, we want to discuss a special yet important case, where the infinite series is reduced to only the first term. In fact, when $G$ is an **Abelian Lie group**, for any $\xi, \hat{\xi} \in \mathfrak{g}$, the Lie bracket $[\xi, \hat{\xi}] = 0$ vanishes identically, and the solution to IVP in (9) reduces to $g(t) = g_0 \exp(\int_0^t \xi(s)\mathrm{d}s)$.

**Theorem 2** (Conditional transition probability for Abelian Lie Group). *Let $G$ be an Abelian Lie group which is isomorphic to $\mathsf{SO}(2)$. In this case, the conditional transition probability can be written explicitly as*

$$p_{t|0}(g_t, \xi_t \mid g_0, \xi_0) = \mathrm{WN}(\operatorname{logm}(g_0^{-1}g_t); \mu_g, \sigma_g^2) \cdot \mathcal{N}(\xi_t; \mu_\xi, \sigma_\xi^2) \quad (10)$$

*where* $\mathrm{WN}(x; \mu, \sigma^2)$ *is the density value of the Wrapped Normal distribution with mean* $\mu$ *and variance* $\sigma^2$, *and evaluated at* $x$. *For explicit expressions of* $\mu_g, \sigma_g^2$ *and* $\mu_\xi, \sigma_\xi^2$ *and the multivariate case formula, please see Appendix C.*

An important example of Abelian Lie group is the 1D torus $\mathbb{T}$ or special orthogonal group $\mathsf{SO}(2)$, and any of their direct product. In this case, we can compute the conditional transition probability $p_{t|0}(g_t, \xi_t)$ exactly due to the capability of solving the IVP exactly. We summarize the results in Theorem 2 and leave the proof in Appendix C. Notice that while Theorem 2 only gives the conditional transition probability for $G \cong \mathsf{SO}(2)$, it can be extended to a multivariate case where $G \cong \mathsf{SO}(2)^k$ since the conditional transition of forward dynamic factorizes over each dimension when conditioned on the initial value. For a detailed discussion of the multivariate case, please see Appendix C.

**Implicit Score Matching**  When $G$ is not Abelian, the conditional transition probability in (10) might not be available. In this case, we resort to another computationally tractable variant of the score-matching loss derived by performing integration by parts, also known as the implicit score-matching objective $\mathcal{J}_{\mathrm{ISM}}$ (Hyvärinen, 2005). In fact, we can connect $\mathcal{J}_{\mathrm{ISM}}$ and $\mathcal{J}_{\mathrm{SM}}$ by,

$$\mathcal{J}_{\mathrm{SM}}(\theta) = \underbrace{\mathbb{E}_{t,p_t}\Big[\big\|s_\theta(g, \xi, t)\big\|^2 + 2\,\mathrm{div}_\xi(s_\theta(g, \xi, t))\Big]}_{\mathcal{J}_{\mathrm{ISM}}(\theta)} + C_2$$

where $C_2$ is a constant independent of $\theta$. Hence, $\mathrm{argmin}_\theta\, \mathcal{J}_{\mathrm{SM}} = \mathrm{argmin}_\theta\, \mathcal{J}_{\mathrm{DSM}}$. To evaluate $\mathcal{J}_{\mathrm{ISM}}$, samples approximated distributed as $p_t$ are generated through simulation of forward dynamic. Computing it also requires evaluating the divergence with respect to $\xi$, which is the trace of the Jacobian. For high dimensional problems, stochastic approximations with Hutchinson's trace estimator (Hutchinson, 1989; Song et al., 2020a) are often employed to improve computational efficiency.

## 2.4  NUMERICAL INTEGRATION AND SCORE PARAMETERIZATION

To either simulate the forward dynamic for generating trajectories used for evaluating implicit score matching objective $\mathcal{J}_{\mathrm{ISM}}$ or sampling from the backward dynamic for generating new samples, we need to integrate the dynamic. To exploit the Euclidean structure of $\xi$ to achieve higher numerical accuracy, we introduce the **Operator Splitting Integrator (OSI)**. Apart from enjoying a better prefactor in terms of numerical errors, such an integrator is also manifold-preserving and projection-free. Details of the integrator can be found in Appendix D, along with a convergence analysis of OSI (Appendix D.3) which is essentially a reproduction of the proof in Kong & Tao (2024).

**Integrating forward dynamic**  In order to numerically integrate the forward dynamic (3), we note that the dynamic can be split into the sum of two much simpler dynamics depicted in (11). This is the approach considered by Kong & Tao (2024).

$$A_g^{\mathcal{F}} : \left\{ \begin{array}{l} \dot{g}_t = g_t \xi_t \\ \mathrm{d}\xi_t = 0\,\mathrm{d}t \end{array} \right. \quad + \quad A_\xi^{\mathcal{F}} : \left\{ \begin{array}{l} \dot{g}_t = 0 \\ \mathrm{d}\xi_t = -\gamma\xi_t\mathrm{d}t + \sqrt{2\gamma}\mathrm{d}W_t^{\mathfrak{g}} \end{array} \right. \tag{11}$$

While the original forward dynamic does not in general have a simple, closed-form solution for non-Abelian groups, the two smaller systems $A_g^{\mathcal{F}}$ and $A_\xi^{\mathcal{F}}$ are essentially linear and both allow exact integration with closed-form solutions. Therefore, instead of directly integrating the forward dynamic, we can integrate $A_g^{\mathcal{F}}$ and $A_\xi^{\mathcal{F}}$ alternatively for each timestep. Another notable property of such integration is that the trajectory of this numerical integration scheme will stay exactly on the manifold $G \times \mathfrak{g}$. This avoids the use of projection operators at the end of each timestep to ensure the iterates stay on the manifold. By performing such a manifold-preserving integration technique, we not only get rid of the inaccuracy caused by projections but also greatly reduce the implementation difficulties since such projections in general do not admit closed-form formulas.

**Integrating backward dynamic**  To perform generative modeling and sample from the backward dynamic, we can either directly work with the stochastic backward dynamic in (7) or its corresponding marginally-equivalent probability flow ODE. We discuss mainly the integrators for the stochastic dynamic and defer the discussion of probability flow ODE to Appendix D. Employing a similar

operator splitting scheme, dynamic (7) can be split into the following two simpler dynamics,

$$A_g^{\mathcal{B}} : \left\{ \begin{array}{l} \dot{g}_t = -g_t \xi_t \\ \mathrm{d}\xi_t = 0\,\mathrm{d}t \end{array} \right. \quad + \quad A_\xi^{\mathcal{B}} : \left\{ \begin{array}{l} \dot{g}_t = 0 \\ \mathrm{d}\xi_t = \gamma \xi_t \mathrm{d}t + 2\gamma s_\theta(g_t, \xi_t, t)\mathrm{d}t + \sqrt{2\gamma}\mathrm{d}W_t^{\mathfrak{g}} \end{array} \right. \tag{12}$$

While $A_g^{\mathcal{B}}$ still allows exact integration and helps preserve the trajectory on the Lie group, $A_\xi^{\mathcal{B}}$ no longer has a closed form solution due to the nonlinearity in $s_\theta$. In this case, we still use exponential integrators to conduct the exact integration of the linear component and discretize the nonlinear component by using a left-point rule, i.e. pretending that $g$ and $\xi$ do not change over a short time $h$. This treatment is beyond the consideration by Kong & Tao (2024) but it is a rather natural extension.

**Score parameterization**  Previous works on manifold generative modeling like RFM (Chen & Lipman, 2024), RDM (Huang et al., 2022), and RSGM (De Bortoli et al., 2022) often require learning a score that belongs to the tangent space at the input, *i.e.*, $s_\theta(g, t) \in T_g G$. This means that the score network at each input $g$ needs to adapt individually to the geometric structure at that point. One thus needs to either write explicitly the $g-$dependent isomorphism between $T_g G$ and $\mathbb{R}^d$ for each $g$, or embed $T_g G$ in the Euclidean space $\mathbb{R}^n$ with $n \gg d$ and apply projections onto $T_g G$ to obtain a valid score. Either way, one needs to handle the geometry of $G$ and/or deal with additional approximation errors and computational costs (e.g., incurred by projections), and learn a hard object in a changing space with structural constraints.

On the other hand, since our approach only needs to approximate the score $\nabla_\xi \log p_t$, which is an element in the Lie algebra $\mathfrak{g}$, we can use a standard Euclidean-valued neural network to universally approximate $s_\theta$. Thanks to the technique of trivialization, we can enjoy the already demonstrated success of score learning in a fixed Euclidean space, where the non-Euclidean effects stemming from the Riemannian geometry are extracted and represented through the left-multiplied $g$ position variable. The need to parameterize the score function in a geometry-dependent space is by-passed, without any approximation in this step. The hardwiring of the geometric structural constraints into the dynamics greatly reduces the implementation difficulty, improves the efficiency of score representation learning, releases the flexibility to choose score parameterization to users, and potentially makes the generative model more data efficient as there is no more need to learn the geometry from data.

## 3  EXPERIMENTAL RESULTS

We will demonstrate accurate generative modeling of Lie group data corresponding to 1) complicated and/or high-dim distribution on torus, 2) protein and RNA structures, 3) sophisticated synthetic datasets on possibly high-dim Special Orthogonal Group, and 4) an ensemble of quantum systems, such as quantum oscillator with a random potential or Random Transverse Field Ising Model (RTFIM), characterized by their time-evolution operators. Details of the dataset and training set-up are discussed in Appendix G.

**Evaluation Methodology:** We adhere to the standard evaluation criterion in Riemannian generative modeling, which is Negative Log Likelihood (NLL). A consistent number of function evaluations is maintained as per prior studies. All datasets were meticulously partitioned into training and testing sets using a 9:1 ratio. Details of NLL estimation procedure are in Appendix F; note that result is not new and only for completeness, but our proof is particularly adapted to Lie group manifolds, intrinsic and independent of the choice of charts and coordinates.

**Complicated and/or High Dimensional Torus Data:**  We start by comparing our model performance with RFM Chen & Lipman (2024) on intricate datasets such as the checkerboard and Pacman on $\mathbb{T}^2$, which are discontinuous and multi-modal. Here, Pacman is a dataset newly curated by us to test generation on torus in challenging situations. It was noted (Lou et al., 2023, Fig.3) that RFM produces less satisfactory results when generating complicated patterns on torus, such as the checkerboard with a size larger than $4 \times 4$. We observed that RFM, although a very strong method, ran into a similar issue when generating Pacman, which is arguably more sophisticated. Figure 3 and Figure 4 show that our model consistently exhibited proficiency in generating intricate patterns within the torus manifold. A scalability study shown in Figure 2 confirmed our method's good scalability to

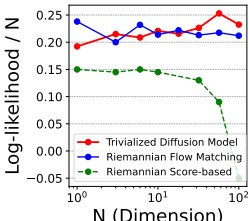

Figure 2: Log likelihood ($\uparrow$) v.s. Dimensions.

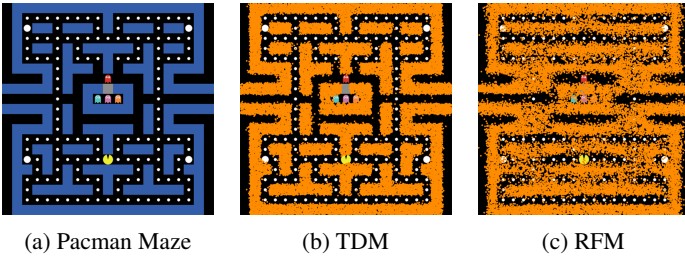

| (a) Pacman Maze | (b) TDM | (c) RFM |

Figure 4: Visualization of Pacman dataset and generated data by TDM on $\mathbb{T}^2$. Pacman maze corresponds to a random variable on $\mathbb{T}^2$ with a complicated distribution corresponding to locations where there is a wall.

high-dimensional cases with minimal degradation in performance (NLL). For the scalability study, we adopted the same setting considered in RFM and compared with its results.

**Protein/RNA Torison Angles on Torus:** We also test on the popular protein Lovell et al. (2003) and RNA Murray et al. (2003) datasets compiled by Huang et al. (2022). These datasets correspond to configurations of macro-molecules represented by torsion angles (hence non-Euclidean), which are 2D or 7D. Results, including generated data of the protein datasets, are presented in Table 1 and Figure 7. The results of RFM were taken from (Chen & Lipman, 2024), where RDM was compared to and results of RSGM were not provided. Notably, our model outperforms the baselines by a significant margin, as evidenced by the visualizations of RNA illustrating the alignment of generated data with ground truth via density plots. The empirical results demonstrating our model's substantial performance gains are possibly rooted in the proposed simulation-free training, high-accuracy sampling, and reduced number of approximations.

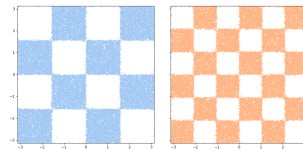

Figure 3: Visualization of Generated data by TDM on $4 \times 4$ and $6 \times 6$ checkerboard.

Table 1: Test NLL ($\downarrow$) over Protein/RNA datasets

| Model | General (2D) | Glycine (2D) | Proline (2D) | Pre-Pro (2D) | RNA (7D) |
|---|---|---|---|---|---|
| **Dataset size** | 138208 | 13283 | 7634 | 6910 | 9478 |
| RDM | $1.04 \pm 0.012$ | $1.97 \pm 0.012$ | $0.12 \pm 0.011$ | $1.24 \pm 0.004$ | $-3.70 \pm 0.592$ |
| RFM | $1.01 \pm 0.025$ | $1.90 \pm 0.055$ | $0.15 \pm 0.027$ | $1.18 \pm 0.055$ | $-5.20 \pm 0.067$ |
| **TDM** | $\mathbf{0.69 \pm 0.14}$ | $\mathbf{1.04 \pm 0.27}$ | $\mathbf{-0.60 \pm 0.15}$ | $\mathbf{0.52 \pm 0.10}$ | $\mathbf{-6.86 \pm 0.46}$ |

| Model | Log likelihood |
|---|---|
| RSGM | $0.20 \pm 0.03$ |
| **TDM** | $\mathbf{0.292 \pm 0.07}$ |

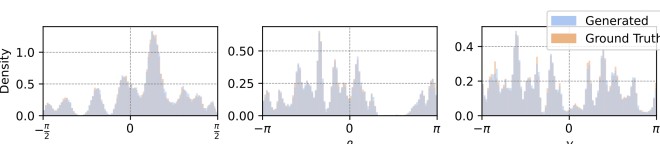

Figure 5: Log likelihood and visualization of generated data for $\mathsf{SO}(3)$ with 32 mixture components.

**Special Orthogonal Group in High Dimensions:** We now evaluate our model's performance on $\mathsf{SO}(n)$ data. Notably, our model is the first reported one to successfully generate beyond $n = 3$. For $\mathsf{SO}(3)$, we generate a difficult mixture distribution in the same way as in (De Bortoli et al., 2022). We also generate data for $\mathsf{SO}(n)$ with $n > 3$ in a similar fashion. With the trivialization technique, we bypass the need to compute the Riemannian logarithm map used in RFM training or eigenfunctions of the heat kernel on $\mathsf{SO}(n)$, which is needed by RSGM (De Bortoli et al., 2022) but in general does not admit a tractable form. The accuracy of our approach can be seen from both the visualization and NLL metric in Figure 5 and Figure 6.

**Learning an Ensemble of Quantum Processes:** Lastly, we experiment with a complex-valued Lie group, the unitary group $\mathsf{U}(n)$. $\mathsf{U}(n)$ holds critical importance in, e.g., high energy physics (Weinberg, 1995) and quantum sciences (Nielsen & Chuang, 2010). Our approach, arguably for the first time, tackles the generative modeling of $\mathsf{U}(n)$ data and manages to scale to nontrivial dimensions. Specifically, how a quantum system evolves is encoded by a unitary operator, i.e. an element in $\mathsf{U}(n)$, and we consider training data corresponding to the time-evolution operators of an ensemble of quantum systems, and aim at generating more quantum systems that are similar to the training data.

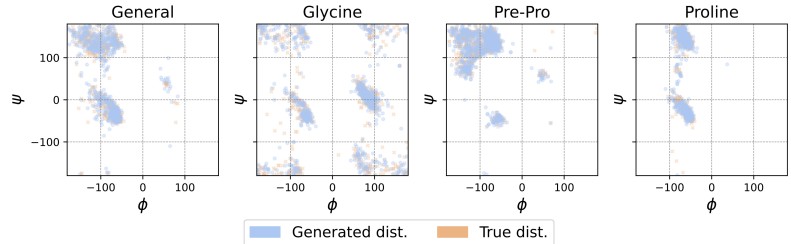

Figure 6: Visualization of Generated $\mathsf{SO}(n)$ data. We scatter plot the marginals of randomly selected dimensions. **Left:** $\mathsf{SO}(4)$. **Middle:** $\mathsf{SO}(6)$. **Right:** $\mathsf{SO}(8)$

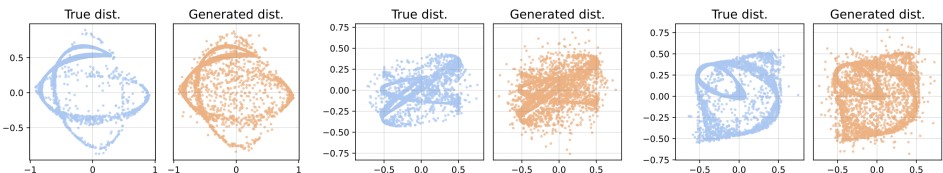

Figure 7: Visualization of Generated Protein Torsion Angle. $\psi$ and $\phi$ are torsion angles on the protein backbone. We scatter plot the ground truth distribution and overlay it with generated torsion angles. The high overlap suggests a good match between the generated distribution and true distribution.

Two examples are tested, respectively quantum oscillators in random potentials, and Transverse Field Ising Model with random couplings and field strength. (Spatial discretization, if needed, of) the time evolution operator of Schrödinger equation for each system gives one $\mathsf{U}(n)$ data point in the training set. Fig.8 provides marginals' scatter plots to showcase the fidelity of our generated distributions, for Quantum Oscillators. Fig.9 is for Transverse Field Ising Model.

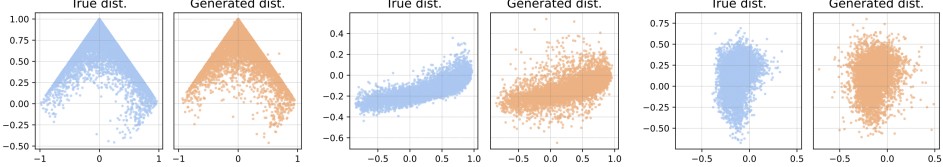

Figure 8: Visualization of Generated Time-evolution Operator of Quantum Oscillator on $\mathsf{U}(n)$. Time-evolution operators are of the form $e^{it\mathcal{H}}$, where $\mathcal{H} = \Delta_h - V_h$, $\Delta_h$ is the discretized Laplacian and $V_h(x) = \frac{1}{2}\omega^2|x-x_0|^2$ is a random potential function. We scatter plot the marginals of randomly selected dimensions. **Left:** $\mathsf{U}(4)$. **Middle:** $\mathsf{U}(6)$. **Right:** $\mathsf{U}(8)$

Figure 9: Visualization of Generated Time-evolution Operator of Transverse Field Ising Model (TFIM) on $\mathsf{U}(n)$. Time-evolution operators are of the form $e^{it\mathcal{H}}$, where $\mathcal{H}$ is the Hamiltonian of the TFIM with a random coupling parameter and field strength. We scatter plot the marginals of randomly selected dimensions. **Left:** 2-qubit TFIM, $\mathsf{U}(4)$. **Middle:** 3-qubit TFIM, $\mathsf{U}(8)$. **Right:** 3-qubit TFIM, $\mathsf{U}(8)$

## 4 CONCLUSION, LIMITATION, AND FUTURE POSSIBILITIES

We propose TDM, an approximation-free diffusion model on Lie groups, by algorithmically introducing trivialized momentum, which turns a manifold problem to a Euclidean-like situation. Compared to existing milestones such as RFM, RSGM and RDM, TDM achieves superior performance on various benchmark datasets thanks to having fewer sources of approximations. However, trivialization strongly leverages group structure and does not directly generalize to general manifolds, although it is possible to extend to homogeneous spaces. Meanwhile, there is potential for improvement in the score parameterization and training, such as by adopting techniques such as preconditioning and exponential moving average tuning (Karras et al., 2022), which will be investigated in the future.

ACKNOWLEDGMENT

We thank Michael Brenner, Valentin De Bortoli, and Yuehaw Khoo for encouragements and inspirations. YZ, LK, and MT are grateful for partial supports by NSF Grant DMS-1847802, Cullen-Peck Scholarship, Emory-GT AI.Humanity Award, and MathWorks Microgrant.

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

# A BACKGROUNDS

**Diffusion Generative Model in Euclidean spaces**: We first review Diffusion Generative Models (sometimes also referred to as Score-based Generative Model, denoising diffusion model, etc.) (Ho et al., 2020; Song et al., 2020b). Here, we adopt the Stochastic Differential Equations (SDE) description (Song et al., 2020b). Given samples of $\mathbb{R}^d$-valued random variable $X_0$ that follows the data distribution $p_0$ which we are interested in, denoising diffusion adopts a forward noising process followed by a backward denoising generation process to generate more samples of $p_0$. The forward process transports the data distribution to a known, easy-to-sample distribution by evolving the initial condition via an SDE,

$$\mathrm{d}X_t = f(X_t, t)\mathrm{d}t + \sqrt{2\gamma(t)}\mathrm{d}W_t, \quad X_0 \sim p_0. \tag{13}$$

In this case, $p_{+\infty}$ will be a standard Gaussian $\mathcal{N}(0, I)$ with appropriate choice of $\gamma(t)$. The backward process then utilizes the time-reversal of the SDE (13) (Anderson, 1982). More precisely, if one considers

$$\mathrm{d}Y_t = \left(- f(Y_t, t) + 2\gamma(t)\nabla \log p_{T-t}(Y_t)\right)\mathrm{d}t + \sqrt{2\gamma(T-t)}\mathrm{d}W_t, \quad Y_0 \sim p_T. \tag{14}$$

Then we have $Y_t \sim p_{T-t}$, i.e. $Y_t = X_{T-t}$ in distribution. In particular, the $T$-time evolution of (14), $Y_T$, will follow the data distribution $p_0$. In practice, one considers evolving the forward dynamics for finite but large time $T$, so that $p_T \approx \mathcal{N}(0, I)$, and then initialize the backward dynamics using $Y_0 \sim \mathcal{N}(0, I)$ and simulate it numerically till $t = T$ to obtain approximate samples of the data distribution. Critically, the score function $s$ needs to be estimated in the forward process.

To do so, the score $\nabla \log p_t$ is often approximated using a neural network $s_\theta$. For linear forward SDE, it is typically trained by minimizing an objective based on denoising score matching (Vincent, 2011), namely

$$\mathbb{E}_t \mathbb{E}_{X_0 \sim p_0} \mathbb{E}_{X_t \sim p_t(\cdot|X_0)} \|s_\theta(X_t, t) - \nabla \log p_t(X_t|X_0)\|^2 \tag{15}$$

where $\nabla \log p_t(X_t|X_0)$ is the conditional score derived from the solution of (13) with a given initial condition.

**Kinetic Langevin dynamics in Euclidean spaces, and CLD**: When Einstein first proposed 'Brownian motion', he actually thought of a mechanical system under additional perturbations from noise and friction (Einstein, 1905). This is now (generalized, formalized, and) known as the kinetic Langevin dynamics (e.g., Nelson, 1967), i.e.

$$\begin{cases} dQ &= M^{-1}P dt \\ dP &= -\gamma P dt - \nabla V(Q) dt + \sigma dW_t \end{cases} \tag{16}$$

which converges, as $t \to \infty$, to a limiting probability distribution $Z^{-1}\exp(-(P^T M^{-1}P/2 + V(Q))/T)dPdQ$ under mild conditions, where $M$ is mass matrix that can be assumed to be $I$ without loss of generality, and $T = \sigma^2/(2\gamma)$ is known as the temperature. If $T$ is fixed and $\gamma \to \infty$, one recovers (in distribution and after time rescaling) overdamped Langevin dynamics

$$dQ = -\nabla V(Q)dt + \sqrt{2T}dW_t.$$

Just like how overdamped Langevin (often with $V$ being quadratic) can be used as the forward process for diffusion generative model, kinetic Langevin can also be used as the forward process. In fact, in a seminal paper, Dockhorn et al. (Dockhorn et al., 2021) used it to smartly bypass the singularity of score function at $t = 0$ when overdamped Langevin is employed as the forward process and data is supported on low dimensional manifolds and called the resulting method CLD. Similar to Equation 14, one can construct the reverse process of Equation 16 as follow,

$$\begin{cases} dQ &= -M^{-1}P dt \\ dP &= \gamma P dt + \nabla V(Q)dt + \sigma^2 \nabla_P \log p(Q, P, t)dt + \sigma dW_t. \end{cases} \tag{17}$$

By endowing $P$ with Gaussian initial condition, $p$ is fully supported in $P$ space, and since the score function only takes the gradient with respect to $P$, it no longer has the aforementioned singularity issue when $t$ tends to zero. This benefits score parameterization and learning.

Finally, let us contrast the trivialized Lie group kinetic Langevin dynamics (used as forward dynamics in this work) with the classical Euclidean kinetic Langevin dynamics by setting $\sigma = \sqrt{2\gamma}$ and $M = 1$:

$$\text{Lie: } \begin{cases} \dot{g} = g\xi, \\ \mathrm{d}\xi = -\gamma\xi\mathrm{d}t - T_g\mathsf{L}_{g^{-1}}(\nabla U(g))\mathrm{d}t + \sqrt{2\gamma}\mathrm{d}W_t^{\mathfrak{g}}, \end{cases} \qquad \text{Euclidean: } \begin{cases} \dot{Q} = P, \\ \mathrm{d}P = -\gamma P\mathrm{d}t - \nabla U(Q)\mathrm{d}t + \sqrt{2\gamma}\mathrm{d}W_t, \end{cases}$$

Note the main difference is the 1st line, i.e. the position dynamics; the 2nd line is identical except for conservative forcing, but that has to be different in the manifold case.

**Lie group:** A *Lie group* is a differentiable manifold that also has a group structure, denoted by $G$. A *Lie algebra* is a vector space with a bilinear, alternating binary operation that satisfies the Jacobi identity, known as the Lie bracket. The tangent space of a Lie group at $e$ (the identity element of the group) is a *Lie algebra*, denoted as $\mathfrak{g} := T_eG$.

## B  TIME REVERSAL FORMULA

In this section, we will prove the time reversal formula stated in Theorem 1. We first introduce an important lemma and calculate the adjoint of the infinitesimal generator of a general diffusion process. We then apply the lemma to derive the Fokker-Planck equation for our process of interest and finish the proof.

**Lemma 1.** *Given $\alpha, \beta : G \times \mathfrak{g} \to \mathfrak{g}$, let $\mathcal{L}$ denote the infinitesimal generator of the following dynamic*

$$\begin{cases} \dot{g} = T_eL_g\alpha(g,\xi)\mathrm{d}t \\ \mathrm{d}\xi = \beta(g,\xi)\mathrm{d}t + \sqrt{2\gamma(t)}\mathrm{d}W_t \end{cases} \tag{18}$$

*The adjoint of $\mathcal{L}$ is given by*

$$\mathcal{L}^*p = -\operatorname{div}_g(pT_eL_g\alpha) - \operatorname{div}_\xi(p\beta) + \gamma(t)\Delta_\xi p$$

*Proof of Lemma 1.* We first write down the infinitesimal generator $\mathcal{L}$ for SDE (18). For any $f \in C_0^2(G \times \mathfrak{g})$, $\mathcal{L}$ is defined as

$$\mathcal{L}f(g,\xi) := \lim_{\delta \to 0} \frac{\mathbb{E}\left[f(g_\delta, \xi_\delta)|(g_0, \xi_0) = (g, \xi)\right] - f(g,\xi)}{\delta}$$
$$= \langle \nabla_g f, T_eL_g\alpha \rangle + \langle \nabla_\xi f, \beta \rangle + \gamma(t)\Delta_\xi f$$

By definition, $\mathcal{L}^* : C^2 \to C^2$ (the adjoint operator of $\mathcal{L}$) satisfies $\int_{G \times \mathfrak{g}} p\mathcal{L}f\mathrm{d}g\mathrm{d}\xi = \int_{G \times \mathfrak{g}} f\mathcal{L}^*p\,\mathrm{d}g\mathrm{d}\xi$ for any $f, p \in C_0^2(G \times \mathfrak{g})$. By the divergence theorem, we have

$$\int_{G \times \mathfrak{g}} p\mathcal{L}f\,\mathrm{d}g\mathrm{d}\xi = \int_{G \times \mathfrak{g}} f\left(-\operatorname{div}_g(pT_eL_g\alpha) - \operatorname{div}_\xi(p\beta) + \gamma(t)\Delta_\xi p\right)\mathrm{d}g\mathrm{d}\xi$$

Here $T_eL_g\xi$ stands for the left-invariant vector filed on $G$ generated by $\xi \in \mathfrak{g}$. As a result, we have

$$\mathcal{L}^*p = -\operatorname{div}_g(pT_eL_g\alpha) - \operatorname{div}_\xi(p\beta) + \gamma(t)\Delta_\xi p$$

$\square$

We are now ready to show that the backward dynamic (5) is the time reversal process of the forward dynamic (1). Fokker-Planck characterizes the evolution of the density of a stochastic process: denote the density at time $t$ as $\rho_t$, we have $\rho_t$ satisfies $\frac{\partial}{\partial t}\rho_t = \mathcal{L}^*\rho_t$. Thm. 1 is proved by comparing the Fokker-Planck equation for Eq. (1) and (5).

*Proof of Theorem 1.* By denoting the density for SDE following the forward dynamic (1) as $p_t$ and the density for SDE following backward dynamic (5) as $\tilde{p}_t$, we only need to prove $p_t \equiv \tilde{p}_{T-t}$.

Using Lemma 1, the Fokker-Planck equation of the forward dynamic Eq. (1) is given by,

$$\frac{\partial}{\partial t} p_t = -\operatorname{div}_g(p_t T_e L_g \xi) + \gamma(t) \operatorname{div}_\xi(p_t \xi) + \gamma(t) \Delta_\xi p_t$$

and the Fokker-Planck equation for Eq. (5) is given by

$$\frac{\partial}{\partial t} \tilde{p}_t = -\operatorname{div}_g(-\tilde{p}_t T_e L_g \xi) - 2\gamma \operatorname{div}_\xi(\tilde{p}_t \nabla_\xi \log \tilde{p}_t) - \gamma(T-t) \operatorname{div}_\xi(\tilde{p}_t \xi) + \gamma(T-t) \Delta_\xi \tilde{p}_t$$

$$= \operatorname{div}_g(\tilde{p}_t T_e L_g \xi) - \gamma(T-t) \operatorname{div}_\xi(\tilde{p}_t \xi) - \gamma(T-t) \Delta_\xi \tilde{p}_t$$

where the last equation holds due to $\operatorname{div}_\xi(\tilde{p}_t \nabla_\xi \log \tilde{p}_t) = \operatorname{div}_\xi(\nabla_\xi \tilde{p}_t) = \Delta_\xi p_t$.
We can calculate the partial derivative of the reversed distribution $\tilde{p}_{T-t}$ with respect to $t$, which gives

$$\frac{\partial}{\partial t} \tilde{p}_{T-t} = -\operatorname{div}_g(\tilde{p}_t T_e L_g \xi) + \gamma(t) \operatorname{div}_\xi(\tilde{p}_t \xi) + \gamma(t) \Delta_\xi \tilde{p}_t$$

Note that this exactly matches the expression for $\frac{\partial}{\partial t} p_t$. Together with the same initial condition $\tilde{p}_0 = p_T$, we deduce that $p_t \equiv \tilde{p}_{T-t}$ for all $t$. $\qquad\square$

## C  DENOISING SCORE MATCHING FOR ABELIAN LIE GROUP

We first state a detailed version of Theorem 2 in the following,

**Corollary 1** (Conditional transition probability for Abelian Lie Group). *Let $G$ be an Abelian Lie group which is isomorphic to $\mathbb{T}$ or $\mathsf{SO}(2)$. In this case, the conditional transition probability can be written explicitly as,*

$$p_{t|0}(g_t, \xi_t \mid g_0, \xi_0) = \mathrm{WN}(\operatorname{logm}(g_0^{-1} g_t); \mu_g, \sigma_g^2) \cdot \mathcal{N}(\xi_t; \mu_\xi, \sigma_\xi^2) \qquad (19)$$

*where $\mathrm{WN}(x; \mu, \sigma^2)$ is the density of the Wrapped Normal distribution with mean $\mu$ and variance $\sigma^2$ evaluate at $x$, $\operatorname{logm}$ is the matrix logarithm with principal root, and $\mu_g, \mu_\xi, \sigma_g^2, \sigma_\xi^2$ are are given by,*

$$\mu_g = \frac{1 - e^{-t}}{1 + e^{-t}}(\xi_t + \xi_0), \ \mu_\xi = e^{-t}\xi_0$$

$$\sigma_g^2 = 2t + \frac{8}{e^t + 1} - 4, \ \sigma_\xi^2 = 1 - e^{-2t}$$

In this section, we will prove Theorem 2 by proving its detailed version in Corollary 1, under the condition that $G$ is an Abelian Lie group which is isomorphic to $\mathbb{T}$ or $\mathsf{SO}(2)$. Note that this allows us to compute the conditional transition probability for $G$ that is also direct product of these Lie groups. The reason is that, for a Lie group $G$ that is a direct product of $\mathbb{T}$ and $\mathsf{SO}(2)$, we can represent an element in $G$ as $(g^1, \ldots, g^k)$ where $g^i \in \mathbb{T}$ or $\mathsf{SO}(2)$. The corresponding Lie Algebra can be represented as $(\xi^1, \ldots, \xi^k)$. For each $1 \le i \le k$, $(g^i, \xi^i)$, they follow the following dynamic,

$$\begin{cases} \dot{g}_t^i = g_t \xi_t^i, \\ \mathrm{d}\xi_t^i = -\gamma \xi_t \mathrm{d}t + \sqrt{2\gamma} \mathrm{d}W_t^i \\ g^i(0) = g_0^i, \ \xi^i(0) = \xi_0^i \end{cases} \qquad (20)$$

Note that this will not create any confusion since $\xi^i \in \mathbb{R}$ and there's no need for another superscript to indicate other elements in $\xi^i$. Moreover, an important consequence of the factorization of the dynamic of $(g, \xi)$ as $k$ independent smaller dynamic is that we can also factorize the conditional transition probability for $g, \xi$ as a product of $k$ conditional transition probability, each computed from $g^i, \xi^i$. This means the following,

$$p_{t|0}(g_t, \xi_t \mid g_0, \xi_0) = \prod_{i=1}^{k} p_{t|0}(g_t^i, \xi_t^i \mid g_0^i, \xi_0^i) \qquad (21)$$

Based on (21), we manage to compute the conditional transition probability of any general connected compact Abelian Lie group, since they are necessarily isomorphic to a power of $\mathbb{T}$ or $\mathsf{SO}(2)$(Kirillov, 2008). Therefore, we just need to compute the conditional transition probability for such a base case, which is stated in Theorem 2.

From now on, we will consider $\gamma = 1$ for simplicity. Generalization of our results to time-dependent is straightforward. Let's stick to the notation that $g_0 \in \mathsf{SO}(2)$, $\xi_0 \in \mathbb{R} \cong \mathfrak{so}(2)$. We slightly abuse the notation in the sense that, when considering the dynamic for $\xi$, we are considering a valid SDE on $\mathbb{R}$, while when we are considering the dynamic for $g$, $\xi$ should be understood as its matrix representation in $\mathfrak{so}(2)$, which is a $2 \times 2$ skew-symmetric matrix. Let also denote $Y_t = \int_0^t \xi_s \mathrm{d}s$ for notational simplicity.

Since $G$ is Abelian, $[\xi, \hat{\xi}] = 0$ for any $\xi, \hat{\xi} \in \mathfrak{g}$, the Magnus series $\Omega_k(t) = 0$ in (9) for $k \geq 2$, and the solution to the IVP can be written explicitly as,

$$
\begin{cases}
g_t = g_0 \operatorname{expm}(Y_t) \\
\xi_t = e^{-t}\xi_0 + \sqrt{2} \int_0^t e^{-(t-s)} \mathrm{d}W_s
\end{cases}
\tag{22}
$$

Notice that $(g_t, \xi_t)$ is a push forward of $(\xi_t, Y_t)$. Therefore, to find the condition transition $p_{t|0}(g_t, \xi_t \mid g_0, \xi_0)$, we first compute the joint distribution of $(\xi_t, Y_t)$ conditioned on the $(g_0, \xi_0)$, and derive the desired conditional transition probability by computing the probability change of variable.

Since $Y_t$ is the time integral of $\xi_t$, $(\xi_t, Y_t)$ is a Gaussian process, with mean and covariance stated in the following Lemma,

**Lemma 2.** *For a given $t$, $(\xi_t, Y_t)$ is distributed according to a bivariate Gaussian,*

$$
\begin{pmatrix} \xi_t \\ Y_t \end{pmatrix} \sim \mathcal{N}\left( \begin{pmatrix} e^{-t}\xi_0 \\ (1-e^{-t})\xi_0 \end{pmatrix}, \begin{pmatrix} 1-e^{-2t} & e^{-2t}(e^t-1)^2 \\ e^{-2t}(e^t-1)^2 & 4e^{-t}-e^{2t}+2t-3 \end{pmatrix} \right)
\tag{23}
$$

*Proof of Lemma 2.* To show that the joint distribution $(\xi_t, Y_t)$ as the desired expression, we just need to compute the mean and variance of $\xi_t$ and $Y_t$ respectively, as well as their covariance. For $\xi_t$,

$$
\mathbb{E}[\xi_t] = \mathbb{E}\left[ e^{-t}\xi_0 + \sqrt{2}\int_0^t e^{-(t-s)}\mathrm{d}W_s \right] = e^{-t}\xi_0
$$

$$
\operatorname{Var}(\xi_t) = \operatorname{Var}\left( \sqrt{2}\int_0^t e^{-(t-s)}\mathrm{d}W_s \right) = 2\int_0^t e^{-2(t-s)}\mathrm{d}s = 1 - e^{-2t}
$$

For $Y_t$, since it's the integration of $\xi_t$, it has the following expression,

$$
\begin{aligned}
Y_t &= \int_0^t e^{-s}\xi_0 \mathrm{d}s + \sqrt{2}\int_0^t \int_0^p e^{-(p-s)}\mathrm{d}W_s \mathrm{d}p \\
&= (1-e^{-t})\xi_0 + \sqrt{2}\int_0^t \int_s^t e^{-(p-s)}\mathrm{d}p \mathrm{d}W_s \\
&= (1-e^{-t})\xi_0 + \sqrt{2}\int_0^t (1-e^{-(t-s)})\mathrm{d}W_s
\end{aligned}
$$

where we use Stochastic Fubini's theorem to exchange the integration order of $\mathrm{d}W_s$ and $\mathrm{d}p$. Therefore, we can compute the mean and variance of $Y_t$,

$$
\mathbb{E}[Y_t] = \mathbb{E}\left[ (1-e^{-t})\xi_0 + \sqrt{2}\int_0^t (1-e^{-(t-s)})\mathrm{d}W_s \right] = (1-e^{-t})\xi_0
$$

$$
\operatorname{Var}(Y_t) = \operatorname{Var}\left( \sqrt{2}\int_0^t (1-e^{-(t-s)})\mathrm{d}W_s \right) = 2\int_0^t (1-e^{-(t-s)})^2 \mathrm{d}s = 4e^{-t} - e^{2t} + 2t - 3
$$

Finally, we need to compute $\mathrm{Cov}(\xi_t, Y_t)$ to complete the proof, where we use Ito's isometry,

$$\mathrm{Cov}(\xi_t, Y_t) = \mathbb{E}\Big[2\int_0^t e^{-(t-s)}\mathrm{d}W_s \cdot \int_0^t (1 - e^{-(t-s)})\mathrm{d}W_s\Big]$$

$$= 2\int_0^t e^{-(t-s)} \cdot (1 - e^{-(t-s)})\mathrm{d}s = e^{-2t}(e^t - 1)^2$$

$\square$

As a corollary of Lemma 2, we can compute the conditional distribution $Y_t|\xi_t$, here we omit the dependence on $\xi_0, g_0$ for simplicity since all probability considered in this section is conditioned on these two value. The conditional distribution between bivariate Gaussian is equivalent to orthogonal projections,

**Corollary 2.** *For a give $t$, $Y_t|\xi_t$ has distribution*

$$Y_t|\xi_t \sim \mathcal{N}\Big(\frac{1 - e^{-t}}{1 + e^{-t}}(\xi_t + \xi_0), 2t + \frac{8}{e^t + 1} - 4\Big) \tag{24}$$

*Proof.* Let $\boldsymbol{\Sigma}$ and $\boldsymbol{\mu}$ denotes the variance matrix and the mean vector of $(\xi_t, Y_t)$. The $Y_t|\xi_t$ has conditional mean and variance given by,

$$\mathbb{E}[Y_t|\xi_t] = \boldsymbol{\mu}_Y + \boldsymbol{\Sigma}_{Y\xi}\boldsymbol{\Sigma}_{\xi\xi}^{-1}(\xi_t - \boldsymbol{\mu}_\xi)$$

$$\mathrm{Var}[Y_t|\xi_t] = \boldsymbol{\Sigma}_{YY} - \boldsymbol{\Sigma}_{Y\xi}\boldsymbol{\Sigma}_{\xi\xi}^{-1}\boldsymbol{\Sigma}_{\xi Y}$$

Plug in the expression for $\boldsymbol{\Sigma}$ and $\boldsymbol{\mu}$, and the expressions simplify to the desired ones. $\square$

We need the distribution of $Y_t|\xi_t$ due to the following factorization of $p_{t|0}(g_t, \xi_t \mid g_0, \xi_0)$,

$$p_{t|0}(g_t, \xi_t \mid g_0, \xi_0) = p_{t|0}(g_t \mid \xi_t, g_0, \xi_0) \cdot p_{t|0}(\xi_t \mid g_0, \xi_0)$$

Here $p_{t|0}(\xi_t \mid g_0, \xi_0)$ is known due to $\xi_t$ being a Gaussian, we need to compute $p_{t|0}(g_t \mid \xi_t, g_0, \xi_0)$, which is a hard object since it's a distribution on the Lie group $G$. However, we can derive its expression by computing the push-forward of $Y_t|\xi_t, g_0, \xi_0$ by the exponential map. The following theorem characterizes such a change of measure given by the exponential map of Lie group as the push-forward,

**Theorem 3** (Theorem 3.1 in Falosi et al. (Falorsi et al., 2019)). *Let $G$ denotes a Lie group and $\mathfrak{g}$ its Lie algebra. Consider a distribution $m$ on $\mathfrak{g}$ with density $r(\xi)$ with respect to the Lebesgue measure on $\mathfrak{g}$, the push-forward of $m$ to $G$, denoted as $\exp_*(m)$ is absolutely continuous with respect to the Haar measure on $G$, with density $p(g)$ given by,*

$$p(g) = \sum_{\xi \in \mathfrak{g}:\, \mathrm{expm}(\xi)=g} r(\xi)|J(\xi)|^{-1}, \quad g \in G,$$

*where $J(\xi) = \det\Big(\sum_{k=0}^{\infty} \frac{(-1)^k}{(k+1)!}(\mathrm{ad}_\xi)^k\Big)$.*
*Moreover, when $G$ is $\mathsf{SO}(2)$ or $\mathbb{T}$, the scenario simplifies to,*

$$J(\xi) = 1, \forall \xi \in \mathfrak{g}$$

$$\{\xi \in \mathfrak{g}:\, \mathrm{expm}(\xi) = g\} = \{\xi \in \mathfrak{g}:\, \xi = \mathrm{logm}(g) \pm 2k\pi, k \in \mathbb{Z}\}$$

Since the exponential map is not injective, computing the density of the push-forwarded measure at $g \in G$ requires summing over density at all the pre-images $\xi$ of $g$ in $\mathfrak{g}$ and weighted by the inverse of the Jacobian $|J(\xi)|$. Fortunately, for our considered case, both the pre-images and the Jacobian can be explicitly characterized and computed.

Applying Theorem 3 to our case, where $r$ is the density of $Y_t | \xi_t, g_0, \xi_0$, we derive the following expression for $p_{t|0}(g_t \mid \xi_t, g_0, \xi_0)$,

$$p_{t|0}(g_t \mid \xi_t, g_0, \xi_0) = \sum_{k=-\infty}^{\infty} p_{Y_t | \xi_t} \left( \mathrm{logm}(g_0^{-1} g_t) + 2k\pi \right) \tag{25}$$

where $p_{Y_t | \xi_t}$ denotes the density of $Y_t | \xi_t$, which is the density of a Gaussian with mean and variance defined in (24). This is also known as the Wrapped Normal distribution (Mardia & Jupp, 2009), its name comes from the fact that the density is generated by "wrapping a distribution" on a circle. Therefore, we denote such a distribution with notation $\mathrm{WN}(x; \mu, \sigma^2)$ denotes such as density evaluated at point $x$, where $\mu, \sigma^2$ is the mean and variance of the normal distribution being wrapped. This finishes the proof of Theorem 2 and Corollary 1.

## D PROBABILITY FLOW ODE AND OPERATOR SPLITTING INTEGRATOR

In this section, we will first discuss in detail the **Operator Splitting Integrator** (OSI) and how they help integrate the forward and backward trivialized kinetic dynamics accurately in a manifold-preserving, projection-free manner. We then introduce the probability flow ODE of the backward dynamic, which is an ODE that is marginally equivalent to dynamic (5). We will also introduce the OSI for the probability flow ODE.

### D.1 OPERATOR SPLITTING INTEGRATOR

In this section, we will demonstrate how OSIs are constructed from the dynamics and the benefits they enjoy. We restrict our attention to first-order numerical integrators in the following discussion. However, such an approach can be generalized and we can indeed craft an OSI with arbitrary order of accuracy by following the approach in Tao and Ohsawa (Tao & Ohsawa, 2020).

**Forward Integrator:** Recall that the forward dynamic (3) can be split into two smaller dynamics,

$$A_g^{\mathcal{F}} : \left\{ \begin{array}{l} \dot{g}_t = g_t \xi_t \\ \mathrm{d}\xi_t = 0 \, \mathrm{d}t \end{array} \right. \quad + \quad A_\xi^{\mathcal{F}} : \left\{ \begin{array}{l} \dot{g}_t = 0 \\ \mathrm{d}\xi_t = -\gamma \xi_t \mathrm{d}t + \sqrt{2\gamma} \mathrm{d}W_t^{\mathfrak{g}} \end{array} \right.$$

While the original dynamic does not admit a simple closed-form solution, $A_g^{\mathcal{F}}$ and $A_\xi^{\mathcal{F}}$ can be solved explicitly as is shown in the following equations,

$$A_g^{\mathcal{F}} : \ g(t) = g(0) \exp\mathrm{m}(t\xi(0)), \ \xi(t) = \xi(0)$$

$$A_\xi^{\mathcal{F}} : \ g(t) = g(0), \xi(t) = \exp(-\gamma t)\xi(0) + \int_0^t \sqrt{2\gamma} \exp(-\gamma(t-s)) \mathrm{d}W_s^{\mathfrak{g}}$$

Therefore, we can integrate $A_g^{\mathcal{F}}$ and $A_\xi^{\mathcal{F}}$ alternatively for each timestep $h$ in order to integrate the original forward dynamic. The detailed algorithm can be found in Algorithm 2.

**Backward Integrator:** Recall that the backward dynamic (12) can be split into two smaller dynamics,

$$A_g^{\mathcal{B}} : \left\{ \begin{array}{l} \dot{g}_t = -g_t \xi_t \\ \mathrm{d}\xi_t = 0 \, \mathrm{d}t \end{array} \right. \quad + \quad A_\xi^{\mathcal{B}} : \left\{ \begin{array}{l} \dot{g}_t = 0 \\ \mathrm{d}\xi_t = \gamma \xi_t \mathrm{d}t + 2\gamma s_\theta(g_t, \xi_t, t) \mathrm{d}t + \sqrt{2\gamma} \mathrm{d}W_t^{\mathfrak{g}} \end{array} \right.$$

Again, we can write out explicit solutions to $A_g^{\mathcal{B}}$ and $A_\xi^{\mathcal{B}}$ in the following equations,

$$A_g^{\mathcal{B}} : \ g(t) = g(0) \exp\mathrm{m}(-t\xi(0)), \ \xi(t) = \xi(0)$$

$$A_\xi^{\mathcal{B}} : \ g(t) = g(0), \xi(t) = \exp(\gamma t)\xi(0) + \int_0^t \sqrt{2\gamma} \exp(\gamma(t-s)) \mathrm{d}W_s^{\mathfrak{g}} + 2\gamma \int_0^t \exp(\gamma(t-s)) s_\theta(g_s, \xi_s, s) \mathrm{d}s$$

Note that, the solution to $A_\xi^{\mathcal{B}}$, though presented in an explicit form, can't be implemented in practice since we can not integrate exactly the neural network $s_\theta$. We employ an approximation here and discretize the nonlinear component $s_\theta$ by using a left-point rule, i.e. pretending that $g$ and $\xi$ do not change over a short time $h$, and still use the exponential integration technique to conduct the exact integration of the rest of the linear dynamic. The detailed algorithm can be found in Algorithm 3.

---

**Algorithm 2** Forward Operator Splitting Integration (**FOSI**)

---

**Require:** step size $h$, total steps $N$, friction constant $\gamma > 0$, initial condition $\bar{g} \in G, \bar{\xi} \in \mathfrak{g}$
1: Set $g_0 = \bar{g}, \xi_0 = \bar{\xi}$
2: **for** $n = 1, \dots, N$ **do**
3:     Sample i.i.d. $\epsilon_{n-1}^i \sim \mathcal{N}(0, 1 - \exp(-2\gamma h))$ for $1 \le i \le \dim g$
4:     $\xi_n^i = \exp(-\gamma h)\xi_{n-1}^i + \epsilon_{n-1}^i$ ▷ Entrywise exponential integration for $\xi$
5:     $g_n = g_{n-1}\,\mathrm{expm}(h\xi_n)$ ▷ Lie group preserving update for $g$
6: **end for**
7: **return** $\{g_k, \xi_k\}_{k=0,\dots,N}$ ▷ Return whole trajectory

---

**Algorithm 3** Backward Operator Splitting Integration (**BSOI**)

---

**Require:** step size $h$, total steps $N$, friction constant $\gamma > 0$, score network $s_\theta$, initial condition $\bar{g} \in G, \bar{\xi} \in \mathfrak{g}$
1: Set $g_0 = \bar{g}, \xi_0 = \bar{\xi}$
2: **for** $n = 1, \dots, N$ **do**
3:     Set $\boldsymbol{s}_{n-1} = s_\theta(g_{n-1}, \xi_{n-1}, (n-1)h)$
4:     Sample i.i.d. $\epsilon_{n-1}^i \sim \mathcal{N}(0, \exp(2\gamma h) - 1)$ for $1 \le i \le \dim \mathfrak{g}$
5:     $\xi_n^i = \exp(\gamma h)\xi_{n-1}^i + 2(\exp(\gamma h) - 1)s_{n-1}^i + \epsilon_{n-1}^i$ ▷ Entrywise exponential integration for $\xi$
6:     $g_n = g_{n-1}\,\mathrm{expm}(-h\xi_n)$ ▷ Lie group preserving update for $g$
7: **end for**
8: **return** $g_N, \xi_N$ ▷ Return final iterate

---

**Advantages of OSI:** The benefits of using an OSI for integration are threefold.

**(1)** The first benefit of the OSI is high numerical accuracy. In both the forward and backward dynamic, the linear component of the dynamic is integrated **exactly** due to the use of the exponential integration technique. This implies that, while OSI is still a first-order method in terms of the order of errors, it enjoys a smaller prefactor thanks to the reduction in error source compared with the Euler–Maruyama method (EM).

**(2)** The second benefit of OSI is that the trajectories generated stay on the manifold $G \times \mathfrak{g}$ for the whole time, $(g(kh), \xi(kh)) \in G \times \mathfrak{g}$ for any $k \ge 0$. If we use the EM scheme to integrate the Riemannian component of the dynamic, which is the $g$ dynamic, we would arrive at iterates $g((k+1)h) = g(kh) + h \cdot g(kh)\xi(kh)$. Note that since $g(kh)\xi(kh)$ is in the tangent space of Lie group G at point $g(kh)$, moving arbitrary short time $h$ along such direction would result in $g((k+1)h) \notin G$. Therefore, to achieve a valid trajectory on $G$, we need to employ a projection $\pi_G$ onto the manifold, which causes additional numerical errors apart from the time discretization error. If employing OSI, we will be free from such a concern of leaving the manifold and also the projection errors.

**(3)** The third benefit of OSI is that the numerical scheme is projection-free. As we have discussed in point (2), EM method does not respect the Riemannian geometry structure of the Lie group and constantly requires the application of projections to achieve valid iterates. Apart from the numerical error, computing such a projection could be problematic. In general, Lie groups live in a nonconvex set, which naturally raises concerns about the existence of the closed-form formula for such projections and more generally, how to implement them in a fast algorithm. For example, $\mathrm{SO}(n)$ is the set of matrices satisfies $\{X \in \mathbb{R}^{n \times n} \mid X^\top X = XX^\top = I_n\}$, which is characterized by nonlinear constraints. Therefore, finding out the projection onto these Lie groups requires heavy work and needs to be investigated on a case-by-case basis. Not to mention the possibility that these projection functions could be difficult to implement and require heavy computational resources, which is certainly not scalable for large-scale applications and high-dimensional tasks. On the contrary, by employing OSI, we can enjoy a projection-free numerical algorithm and reduce the complexity of both training and generation.

## D.2 PROBABILITY FLOW ODE

In this section, we will introduce the OSI for probability flow ODE. We recall that the probability flow ODE is given by,

$$\begin{cases} \dot{g}_t = -g_t \xi_t, \\ \mathrm{d}\xi_t = \gamma(T-t)\xi_t \mathrm{d}t + \gamma \nabla_\xi \log p_{T-t}(g_t, \xi_t) \mathrm{d}t \end{cases}$$

Similar to the SDE setting, this can be split into two smaller dynamics,

$$A_g^\mathcal{P} : \left\{ \begin{array}{l} \dot{g}_t = -g_t \xi_t \\ \mathrm{d}\xi_t = 0\,\mathrm{d}t \end{array} \right. + A_\xi^\mathcal{P} : \left\{ \begin{array}{l} \dot{g}_t = 0 \\ \mathrm{d}\xi_t = \gamma \xi_t \mathrm{d}t + \gamma s_\theta(g_t, \xi_t, t)\mathrm{d}t \end{array} \right.$$

Similar to the Backward Operator Splitting Integrator (BSOI), we employ an approximation of the $\xi$ dynamic, discretize the nonlinear component $s_\theta$ by using a left-point rule, i.e. pretending that $g$ and $\xi$ do not change over a short time $h$, and use the exponential integration technique to conduct the exact integration of the rest of the linear dynamic. The details can be found in Algorithm 4.

---

**Algorithm 4** Probability Flow ODE

---

**Require:** step size $h$, total steps $N$, friction constant $\gamma > 0$, score network $s_\theta$, initial condition
$\bar{g} \in G, \bar{\xi} \in \mathfrak{g}$
1: Set $g_0 = \bar{g}, \xi_0 = \bar{\xi}$
2: **for** $n = 1, \ldots, N$ **do**
3:     Set $\boldsymbol{s}_{n-1} = s_\theta(g_{n-1}, \xi_{n-1}, (n-1)h)$
4:     $\xi_n^i = \exp(\gamma h)\xi_{n-1}^i + (\exp(\gamma h) - 1)s_{n-1}^i$          ▷ Entrywise exponential integration for $\xi$
5:     $g_n = g_{n-1} \mathrm{expm}(-h\xi_n)$          ▷ Lie group preserving update for $g$
6: **end for**
7: **return** $g_N, \xi_N$          ▷ Return final iterate

---

## D.3 ERROR ANALYSIS OF OPERATOR SPLITTING INTEGRATOR ON LIE GROUPS

In this section, we reproduce the error analysis for OSI on Lie groups provided in the work of Kong & Tao (2024). Such an analysis justifies the convergence of OSI, which we provide just to be self-contained. For technical details of the proof, please kindly refer to the original paper of Kong & Tao (2024).

In this section, we consider the following general form of OSI update,

$$\begin{cases} \tilde{g}_{k+1} = \tilde{g}_k \exp(h\tilde{\xi}_{k+1}) \\ \tilde{\xi}_{k+1} = \exp(-\gamma h)\tilde{\xi}_k - \frac{1-\exp(-\gamma h)}{\gamma} T_g L_{g^{-1}} \nabla U(\tilde{g}_k) + \sqrt{2\gamma} \int_{kh}^{(k+1)h} \exp(-\gamma(h-s))\mathrm{d}W_s \end{cases} \tag{26}$$

where $(\tilde{g}_k, \tilde{\xi}_k)$ denotes the points generated by discrete numerical scheme. We note that by choosing carefully the potential function $U$, this OSI update can correspond to both the forward operator splitting integration in Algorithm 2 and the backward operator splitting integration in Algorithm 3. Therefore, providing a general error and convergence analysis of equation (26) covers the situation of using OSI on the simulation of both the forward noising process and the backward sampling process. Following the same setting as in Kong & Tao (2024), we introduce the following assumption on the regularity of potential function $U$,

**Assumption 1.** *For the Lie group $G$, we assume it is finite-dimensional, connected, and compact. For the potential $U$, We assume it is $L$-smooth under the Riemannian metric in Lemma 3 in Kong & Tao (2024), i.e., there exist constants $L \in (0, \infty)$, s.t.*

$$\left\| T_g L_{g^{-1}} \nabla U(g) - T_{\hat{g}} L_{\hat{g}^{-1}} \nabla U(\hat{g}) \right\| \le L d(g, \hat{g}) \quad \forall g, \hat{g} \in G$$

We further recall that $\pi_0$ is the initial distribution and $\pi^*$ is the stationary distribution. Under these assumptions, we have the following convergence analysis for the OSI:

**Theorem 4** (Nonasymptotic error bound for OSI). *If the initial condition $(\tilde{g}_0, \tilde{\xi}_0) \sim \pi_0$ satisfies $W_\rho(\pi_0, \pi_*) < \infty$ and $\pi_0$ is absolute w.r.t. $\mathrm{d}g\mathrm{d}\xi$, then $\forall k = 1, 2, \ldots$, the density of scheme Eq. (26)*

*starting from $\pi_0$ has the following $W_2$ distance from the target distribution:*

$$W_2(\tilde{\pi}_k, \pi_*) \le C_\rho \left( e^{-ckh} W_\rho(\pi_0, \pi_*) + \frac{E(h)}{1 - \exp(-ch)} \right)$$

*where $E(h) = \mathcal{O}(h^{\frac{3}{2}})$. Note this holds $\forall h > 0$, but $E(h)$ can grow exponentially. $W_\rho$ is the Wasserstein distance induced by a semi-distance $\rho$ (given explicitly in Eq. 7 in Kong & Tao (2024)) and $C_\rho$ is a constant.*

Theorem 4 quantifies the distance between the resulted sampled distribution using OSI and the true distribution in terms of $W_2$ distance. We notice that the upper bound on this Wasserstein shrinks to $0$ as timestep $h$ tends to $0$, suggesting the convergence of OSI under the general setting.

## E    EVOLUTION OF KL DIVERGENCE

In this section, we will discuss the effectiveness of likelihood training in terms of learning the correct data distribution. We will show that the KL divergence between the true data distribution $p_0$ and the learned data distribution can be bounded by accumulated score-matching errors up to an additional discrepancy error caused by a mismatch in the initial condition.

Recall that $\mathbf{X_t} = (g_t, \xi_t)$ is the trajectory of the forward dynamics (3), with $\mathbf{X}_t$ admitting a smooth density $p_t(g_t, \xi_t)$ with respect to the product of Haar measure on $G$ and Lebesgue measure on $\mathfrak{g}$. Let's denote $(q_t)_{t \in [0,T]} = (p_{T-t})_{t \in [0,T]}$. Note that by construction, $q_0 = p_T \approx \pi^*$ when $T$ is large, where $\pi^*$ is the invariant distribution of the forward dynamic (3). Also, $q_T = p_0$ is the initial condition for the forward dynamic, which in practice is the (partially unknown) joint data distribution on $g$ and $\xi$. Recall that we have denoted the sequence of probability distribution $q_t^\theta$ as the density of $\mathcal{L}(\mathbf{Y}_t^\theta)$ with respect to the reference measure, where $\mathbf{Y}_t^\theta$ is the trajectory of the learned backward dynamic in (7).

We have Theorem 5 regarding the KL divergence between the learnt data distribution $q_T^\theta$ and the true data distribution $p_0$,

> **Theorem 5.** *Let $\mathbf{Y}_t^\theta$ be the trajectory of the learnt backward dynamic (7) under initial condition $\mathbf{Y}_0^\theta = \pi_*$, $\mathbf{Y}_t^\theta$ has density $(q_t^\theta)_{t \in [0,T]}$. When the score is given by $s_\theta(g, \xi, t) := \nabla_\xi \log \hat{q}_{T-t}(g, \xi, t)$, where $(\hat{q}_t)_{t \in [0,T]}$ is the density of the forward dynamic (3) under initial condition $\hat{q}_0$ and satisfies $\hat{q}_T = \pi_*$. We then have*
>
> $$D_{\mathrm{KL}}\left(p_0 \parallel q_T^\theta\right) = \int_0^T \int_{G \times \mathfrak{g}} p_{T-t}(g, \xi) \|\nabla_\xi \log p_{T-t}(g, \xi) - s_\theta(g, \xi, t)\|^2 \mathrm{d}g \mathrm{d}\xi \mathrm{d}t + D_{\mathrm{KL}}(p_T \parallel \pi_*)$$

In order to prove Theorem 5, we need the following Lemma that characterizes the time derivative of the KL divergence between two sequences of probability distributions that corresponds to the time marginal of the same SDE with different initial conditions.

> **Lemma 3.** *Given $p_0, q_0$ two distributions on $G \times \mathfrak{g}$. We denote the sequence of probability distributions $(p_t)_{t \ge 0}$ and $(q_t)_{t \ge 0}$ the marginals of the forward dynamic (3) with initial conditions $p_0$ and $q_0$ respectively. Then, $p_t$ and $q_t$ satisfies*
>
> $$\frac{\partial}{\partial t} D_{\mathrm{KL}}(p_t \parallel q_t) = \int_{G \times \mathfrak{g}} \gamma(t) p_t \|\nabla_\xi \log p_t - \nabla_\xi \log q_t\|^2 \mathrm{d}g \mathrm{d}\xi$$
>
> *where $\nabla_\xi$ is the gradient w.r.t. $\xi$.*

Lemma 3 relates the time derivative of KL divergence between two distributions to the difference in their score integrated over the manifold $G \times g$. To prove this lemma, we need to derive the Fokker-Planck equation for $p_t$ and $q_t$ respectively.

*Proof of Lemma 3.* We prove a general case and consider the general form of forward dynamic described in (13). The evolution of $p_t$ and $q_t$ can be characterized by following Fokker-Plank equations,

$$\frac{\partial}{\partial t}p_t = \mathcal{L}^{*,p}p_t = -\operatorname{div}_g(p_t T_e L_g \xi) + \gamma(t)\left(\operatorname{div}_\xi(p_t \xi) + \Delta_\xi p_t\right)$$

$$\frac{\partial}{\partial t}q_t = \mathcal{L}^{*,q}q_t = -\operatorname{div}_g(q_t T_e L_g \xi) + \gamma(t)\left(\operatorname{div}_\xi(q_t \xi) + \Delta_\xi q_t\right)$$

where $\operatorname{div}_g$ is the divergence of vector fields on $G$ under the left-invariant metric we choose, $\operatorname{div}_\xi$ is the divergence in $\mathfrak{g}$ and $\Delta_\xi$ is the Laplace operator on $\mathfrak{g}$. They are well-defined since $\mathfrak{g}$ is a linear space. Consequently, we can evaluate the time derivative of KL divergence between $p_t$ and $q_t$, where the integration is performed over $G \times g$ unless specifically noted,

$$\frac{\partial}{\partial t}D_{\mathrm{KL}}(p_t \parallel q_t) = \frac{\partial}{\partial t}\left(\int p_t \log \frac{p_t}{q_t}\mathrm{d}g\mathrm{d}\xi\right)$$

$$= \int \frac{\partial p_t}{\partial t}\log \frac{p_t}{q_t}\,\mathrm{d}g\mathrm{d}\xi - \int \frac{\partial q_t}{\partial t}\frac{p_t}{q_t}\,\mathrm{d}g\mathrm{d}\xi$$

$$= \int \left(\log \frac{p_t}{q_t}\right)\mathcal{L}^{*,p}p_t\mathrm{d}g\mathrm{d}\xi - \int \left(\frac{p_t}{q_t}\right)\mathcal{L}^{*,q}q_t\mathrm{d}g\mathrm{d}\xi$$

Using the divergence theorem, we have for any smooth function $f : G \times \mathfrak{g} \to \mathbb{R}$, we have

$$\int f \cdot \mathcal{L}^{*,p}p_t\mathrm{d}g\mathrm{d}\xi = \int \langle \nabla_g f, p_t T_e L_g \xi \rangle + \langle \nabla_\xi f, p_t \gamma(t)\left(\xi + \nabla_\xi \log p_t\right)\rangle\mathrm{d}g\mathrm{d}\xi$$

Similar results holds for $\mathcal{L}^{*,q}$. As a consequence, applying the previous equation with $f = \log \frac{p_t}{q_t}$ and $f = \frac{p_t}{q_t}$ respectively, we can write,

$$\frac{\partial}{\partial t}D_{\mathrm{KL}}(p_t \parallel q_t) = \int \left\langle \nabla_g \log \frac{p_t}{q_t}, p_t T_e L_g \xi \right\rangle + \left\langle \nabla_\xi \log \frac{p_t}{q_t}, p_t \gamma(t)\left(\xi + \nabla_\xi \log p_t\right)\right\rangle\mathrm{d}g\mathrm{d}\xi$$

$$- \int \left\langle \nabla_g \frac{p_t}{q_t}, q_t T_e L_g \xi \right\rangle - \left\langle \nabla_\xi \frac{p_t}{q_t}, q_t \gamma(t)\left(\xi + \nabla_\xi \log q_t\right)\right\rangle\mathrm{d}g\mathrm{d}\xi$$

$$= \int \left\langle \nabla_g \frac{p_t}{q_t}, q_t T_e L_g \xi \right\rangle + \left\langle \nabla_\xi \frac{p_t}{q_t}, q_t \gamma(t)\left(\xi + \nabla_\xi \log p_t\right)\right\rangle\mathrm{d}g\mathrm{d}\xi$$

$$- \int \left\langle \nabla_g \frac{p_t}{q_t}, q_t T_e L_g \xi \right\rangle - \left\langle \nabla_\xi \frac{p_t}{q_t}, q_t \gamma(t)\left(\xi + \nabla_\xi \log q_t\right)\right\rangle\mathrm{d}g\mathrm{d}\xi$$

$$= \int \gamma(t)\left\langle \nabla_\xi \frac{p_t}{q_t}, q_t \left(\nabla_\xi \log p_t - \nabla_\xi \log q_t\right)\right\rangle\mathrm{d}g\mathrm{d}\xi$$

$$= \int \gamma(t)p_t\left\|\nabla_\xi \log p_t - \nabla_\xi \log q_t\right\|^2\mathrm{d}g\mathrm{d}\xi$$

This finishes the proof of the desired Lemma. $\qquad\square$

We are now ready to present proof for Theorem 5. Note that under the conditions on $s_\theta$, in fact $q_t^\theta = \hat{q}_{T-t}$. We just need to apply Lemma 3 between $p_t$ and $\hat{q}_t$ to conclude.

*Proof of Thm. 5.* Lemma 3 gives

$$D_{\mathrm{KL}}\left(p_0 \parallel q_T^\theta\right) = D_{\mathrm{KL}}\left(p_0 \parallel \hat{q}_0\right)$$

$$= D_{\mathrm{KL}}\left(p_T \parallel \hat{q}_T\right) + \int_0^T \frac{\partial}{\partial t}D_{\mathrm{KL}}\left(p_t \parallel \hat{q}_t\right)\mathrm{d}t$$

$$= D_{\mathrm{KL}}\left(p_T \parallel \hat{q}_T\right) + \int_0^T \int_{G\times\mathfrak{g}} \gamma(t)p_t\left\|\nabla_\xi \log p_t - \nabla_\xi \log \hat{q}_t\right\|^2\mathrm{d}g\mathrm{d}\xi\mathrm{d}t$$

Using the condition $s_\theta(g, \xi, t) := \nabla_\xi \log \hat{q}_{T-t}(g, \xi, t)$, and $\hat{q}_T = \pi^*$, with the choice $\gamma(t) = 1$, we arrived at the following equation,

$$D_{\mathrm{KL}}\left(p_0 \parallel q_T^\theta\right) = \int_0^T \int_{G \times \mathfrak{g}} p_{T-t}(g, \xi)\|\nabla_\xi \log p_{T-t}(g, \xi) - s_\theta(g, \xi, t)\|^2 \mathrm{d}g\mathrm{d}\xi\mathrm{d}t + D_{\mathrm{KL}}(p_T \parallel \pi_*)$$

$\square$

# F    NLL ESTIMATION WITH INTRINSIC PROOF

In this section, we provide an intrinsic proof of the instantaneous change of variables on a general manifold, which does not depend on local charts in the proof or the formula. While we are now aware that the results are not new and has been discussed in (Chen et al., 2018; Lou et al., 2020; Falorsi & Forré, 2020; Chen & Lipman, 2024), we still provide proof for a self-consistency.

Let $z$ be a random variable whose range is $\mathcal{M}$ and denote its density as $p_0 \in \mathcal{C}(\mathcal{M})$. Given a smooth time-dependent vector field $X(t, \cdot)$, i.e.,$X(t, \cdot) \in \Gamma^\infty(T\mathcal{M})$ for any $t \in [0, T]$[1]. We consider the push forward map generated by the flow of $X$, i.e., $f_s^t : \mathcal{M} \to \mathcal{M}$ satisfies

$$\frac{d}{ds} f_s^t(x) = X(s, f_s^t(x)), \quad \forall x \in \mathcal{M}, 0 \leq s \leq t \leq T$$

with initial condition $f_s^s$ is the identity map for any $s$. We define $p_t$ as the density of $f_0^t(z)$. Then we have the following theorem,

**Theorem 6.** *[Instaneous Change of Variables on Manifold] Consider $p : \mathbb{R} \times \mathcal{M} \to \mathbb{R}$, such that $p_t = p(t, \cdot)$ is the density of $z(t)$, where $z(t)$ is the random variable defined by $z$ pushforward along $X$ for time $t$. Then we have*

$$\frac{d}{dt} \log p(t, f_0^t(x)) = -\operatorname{div} X(t, f_0^t(x)), \qquad \forall x \in \mathcal{M}$$

We first review a standard result (see e.g., Ross et al. (2023)) Lemma 4, and then provide a proof for Theorem 6. The following Lemma 4 describes the relationship between the density of the push-forward as well as the determinant of the differential and corresponding points. We will use this result heavily in our proof.

**Lemma 4.** *For any diffeomorphism $f : \mathcal{M} \to \mathcal{M}$, we have*

$$f_\# p(f(x)) = p(x) \left(\det df(x)\right), \quad \forall x \in \mathcal{M}$$

*$f_\#$ is the push-forward density. On the right-hand side, $df(x) : T_x\mathcal{M} \to T_{f(x)}\mathcal{M}$, denoting the differential of $f$, is a linear map. Consequently, $\det df$ is well-defined and independent of the choice of coordinate.*

With Lemma 4, we are ready to present our proof for Theorem 6.

*Proof of Thm. 6.* In this proof, we use the shorthand notation $x_t := f_0^t(x)$, which also induces $x_0 = x$. Lemma 4 gives

$$\frac{\mathrm{d}}{\mathrm{d}t} \log p(t, f_0^t(x)) = \frac{\mathrm{d}}{\mathrm{d}} \log \left[(f_0^t)_\sharp p\left(f_0^t(x)\right)\right]$$

$$= \frac{\mathrm{d}}{\mathrm{d}t} \log \left[p_0(x) \det df_0^t(x)\right]$$

$$= \frac{\mathrm{d}}{\mathrm{d}t} \log \left[\det df_0^t(x)\right]$$

---

[1]$\Gamma^\infty(T\mathcal{M})$ denotes the set of all smooth vector fields on $\mathcal{M}$

Since $f$ is the pushforward map, it has the semi-group structure, i.e., $f_{t_1}^{t_2} \circ f_{t_2}^{t_3} = f_{t_1}^{t_3}$ for any $t_1 \leq t_2 \leq t_3$, which gives $\det df_{t_1}^{t_2} \cdot \det df_{t_2}^{t_3} = \det df_{t_1}^{t_3}$, and immediately

$$
\frac{\mathrm{d}}{\mathrm{d}t} \log \left[ \det df_0^t \right] = \lim_{\epsilon \to 0^+} \frac{1}{\epsilon} \left( \log \left[ \det df_0^{t+\epsilon}(x) \right] - \log \left[ \det df_0^t(x) \right] \right)
$$

$$
= \lim_{\epsilon \to 0^+} \frac{1}{\epsilon} \log \left[ \det df_t^{t+\epsilon}(x_t) \right]
$$

Consequently,

$$
\frac{\mathrm{d}}{\mathrm{d}t} \log p(t, f_0^t(x)) = -\frac{\partial}{\partial \epsilon} \log \left| \det df_t^{t+\epsilon}(x_t) \right| \Big|_{\epsilon=0}
$$

$$
= -\frac{\frac{\partial}{\partial \epsilon} \left| \det df_t^{t+\epsilon}(x_t) \right| \Big|_{\epsilon=0}}{\left| \det df_t^{t+\epsilon}(x_t) \right| \Big|_{\epsilon=0}}
$$

$$
= -\frac{\partial}{\partial \epsilon} \left| \det df_t^{t+\epsilon}(x_t) \right| \Big|_{\epsilon=0}
$$

where we use $\frac{\partial}{\partial \epsilon}$ to denote the right derivative, and the last equation is because of $f_t^t$ is identity. Jacobi's formula gives $\frac{d}{\mathrm{d}} \det(A) = \mathrm{tr}\left(\dot{A}\right)$ at $A = I$, which tells

$$
\frac{d}{dt} \log p(t, f_0^t(x)) = -\mathrm{tr}\left( \frac{\partial}{\partial \epsilon} df_t^{t+\epsilon}(x_t) \right) \Big|_{\epsilon=0}
$$

Before we proceed, we define two sets of vector fields, $\{E_i\}_{i=1}^d$ and $\{Y_i\}_{i=1}^d$. $\{E_i\}_{i=0}^d$ is defined as a set of smooth coordinate frame, defined on the whole manifold $\mathcal{M}$. $\{Y_i\}_{i=1}^d$ is a set of vector fields along $x_t$ generated by the push forward map $f_t^{t+\epsilon}$, i.e., $df_t^{t+\epsilon}$ is a map from $T_{x_t}\mathcal{M}$ to $T_{x_{t+\epsilon}}\mathcal{M}$, and $Y_i$ satisfies

$$
df_t^{t+\epsilon}(Y_i(x_{t+\epsilon})) = Y_i(x_{t+\epsilon}), \forall i = 1, \ldots, d, \forall t < t + \epsilon \leq T
$$

with constraint $Y_i(x_t) = E_i(x_t)$. Note that $Y_i$ is defined only along the curve $x_t$ for $t \in [0, T]$.

Since we are considering a push forward map $f$ along a time-dependent vector field $X(t, \cdot) \in \Gamma^\infty(T\mathcal{M})$, we consider to make it time-independent by considering the problem in a new space $\tilde{\mathcal{M}} := \mathbb{R} \times \mathcal{M}$, and a new time-independent vector field $\tilde{X} \in \Gamma^\infty(T\tilde{\mathcal{M}})$ defined by

$$
\tilde{X}(t, x) = (1, X(t, X)), \quad t \in [0, T], x \in \mathcal{M}
$$

Since $x_t$ is the integral curve generated by $X$, the integral curve of $\tilde{X}$ with initial condition $(0, x)$ is given by $\tilde{x}_t = (t, x_t)$, i.e., $\tilde{x}_t$ satisfies $\dot{\tilde{x}}_t = X(\tilde{x}_t)$.

Both $\{E_i\}_{i=1}^d$ and $\{Y_i\}_{i=1}^d$ can be extended to $\tilde{\mathcal{M}}$. For $\{E_i\}_{i=1}^d$, this is defined by $\tilde{E}_0 \equiv (1, 0)$ and $\tilde{E}_i = (0, E_i)$. Similarly, for $\{Y_i\}_{i=1}^d$, this is defined by $\tilde{Y}_i(\tilde{x}_t) = (0, Y_i(x_t))$ and $\tilde{Y}_0 \equiv (1, 0)$.

By choosing an arbitrary linear connection $\nabla$ on $\mathcal{M}$, we can also extend $\nabla$ to $\tilde{\mathcal{M}}$, and we denote such induced linear connection as $\tilde{\nabla}$ (see e.g., Do Carmo & Flaherty Francis, 1992, Ex. 1 in Chap. 6). Since $f$ is the pushforward generated by $X$, $\frac{\partial}{\partial \epsilon} f_t^{t+\epsilon}|_{\epsilon=0} = X(t, \cdot)$, and we have

$$
\frac{d}{dt} \log p(t, f_0^t(x)) = -\sum_{i=1}^d \frac{\partial}{\partial \epsilon} \langle df_t^{t+\epsilon}(E(x_t)), E_i(f_t^{t+\epsilon}(x_t)) \rangle|_{\epsilon=0}
$$

$$
= -\sum_{i=1}^d \frac{\partial}{\partial \epsilon} \langle Y_i(f_t^{t+\epsilon}(x_t)), E_i(f_t^{t+\epsilon}(x_t)) \rangle|_{\epsilon=0}
$$

$$
= -\sum_{i=1}^d \tilde{\nabla}_{\tilde{X}} \langle \tilde{Y}_i, \tilde{E}_i \rangle|_{\tilde{x}_t}
$$

$$
= -\sum_{i=1}^d \nabla_X \langle Y_i, E_i \rangle|_{x_t}
$$

We choose $\nabla$ to be the Levi-civita connection. Because it is compatible with the metric, we have

$$\frac{d}{dt} \log p(t, f_0^t(x)) = -\sum_{i=1}^{d} \langle \nabla_X Y_i, E_i \rangle + \langle Y_i, \nabla_X E_i \rangle$$

By the definition of Lie derivative, we have $\mathcal{L}_X Y_i \equiv 0$. Together with the constraint $Y_i(x_t) = E_i(x_t)$, we have $\nabla_X Y_i = \nabla_{Y_i} X = \nabla_{E_i} X$ at $x_t$, which gives

$$\frac{d}{dt} \log p(t, f_0^t(x)) = -\sum_{i=1}^{d} \langle \nabla_{E_i} X, E_i \rangle + \langle E_i, \nabla_X E_i \rangle$$

Now we show $\langle E_i, \nabla_X E_i \rangle = 0$ using local coordinates:

$$\langle E_i, \nabla_X E_i \rangle = \sum X_i \Gamma_{ij}^j = \sum X_i \frac{\partial}{\partial e_i} \ln \sqrt{|g|}$$

where $\Gamma$'s are Christoffel symbols corresponding to the local coordinates $E_i$, defined by $\Gamma_{ij}^k := \langle \nabla_{E_i} E_j, E_k \rangle$. Due to our choice of $E_i$ as orthonormal frames, we have $|g| \equiv 1$ and $\langle E_i, \nabla_X E_i \rangle = 0$ for all $i$.

By simplifying the expression, we arrive at the following desired results, which finishes our proof.

$$\frac{d}{dt} \log p(t, f_0^t(x)) = -\sum_{i=1}^{d} \langle \nabla_{E_i} X, E_i \rangle = -\operatorname{div} X(t, f_0^t(x))$$

$\square$

We can choose $E_i$ as the left-invariant vector fields generated by $e_i$, i.e., $E_i(g) = T_e L_g e_i$, where $e_i \in \mathfrak{g}$ is a set of orthonormal basis. As a corollary, we can compute dynamic for the log probability as the following,

> **Corollary 3** (NLL estimation on Lie group with left-invariant metric). *For SDE in Eq. (1), the time-dependent vector field is given by*
>
> $$X(g, \xi) = (T_e L_g \xi, -\gamma \xi + \beta(t, g, \xi))$$
>
> *Using the fact that* $\operatorname{div}_g(T_e L_g \xi) = 0$*, we have*
>
> $$\frac{d}{dt} \log p(t, f_0^t(x)) = \sum_{i=1}^{d} \left( -\gamma + \frac{\partial}{\partial \xi_i} \beta_i \right)$$

## G   TRAINING SET-UP, DATASET

### G.1   HYPERPARAMETERS

**Hardware:** All the experiments are running on one RTX TITAN, one RTX 3090 and one 4090.

**Architectural Framework:** We employed the score function $s_\theta(g_t, \xi_t, t; \theta)$ parameterized by the same network architecture as outlined in the CLD paper Dockhorn et al. (2021), albeit with varying parameter counts for each task. Specifically, we consider the following architectures. We represent $g_t$ and $x_t$ as real vectors and embed them into latent vectors of hidden dimension $D$ respectively with trainable MLPs, where the latent vectors are denoted as $g_{hid}, \xi_{hid}$. We embed time $t$ into the sinusoid embedding of dimension $D$ as well, denoted as $t_{hid}$. The score network output is given by

$$s_\theta = \text{MLP-SKIP}(\text{GN}(g_{hid} + \xi_{hid} + t_{hid}))$$

where GN is the group norm operation, and MLP-SKIP is a $k$-layer MLP with skip connections. We use SiLU as the activation function for all the MLPs used in the neural network.

For low dimensional experiments such as Torus, $\mathsf{SO}(3)$, we set $D = 256$. For other experiments, we set $D = 512$. We choose varied $k$ based on problem difficulties, ranging from $k = 3$ to $k = 5$.

**Training Hyperparameters:** Throughout our experiments, we maintained the diffusion coefficient $\gamma(t)$ constant at 1, while the total time horizon $T$ varied depending on the task, with a good choice ranging from $T = 5$ to $T = 15$. We use AdamW optimizer to train the neural networks with an initial learning rate of $5 \times 10^{-4}$ with a cosine annealing learning rate scheduler. We train for at most $200k$ iterations with a batch size of $1024$ for each task, and we observe that the model usually converges within $100k$ iterations.

## G.2 Dataset Preparation

**Protein and RNA Torsion Angles:** We access the dataset prepared by Huang et al. (Huang et al., 2022) from the repository of (Chen & Lipman, 2024). We further post-processed it and transformed the data into valid elements of $\mathbb{T}^n$.

**Pacman:** We take the maze of the classic video game Pacman and extract all the pixel coordinates from the image that corresponds to the maze. We post-processed it and transformed the data into valid elements of $\mathbb{T}^2$.

**Special Orthogonal group** $\mathsf{SO}(n)$**:** For $\mathsf{SO}$, we followed the same procedure as the one described in (De Bortoli et al., 2022) and generate a Gaussian Mixture with 32 components, uniformly random mean and variance. For $n > 3$, we follow a similar procedure with a reduced number of mixture components.

**Unitary group** $\mathsf{U}(n)$**:** We considered the unitary group data of the form $\mathrm{expm}(-it\mathcal{H})$, which is the time evolution operator of the following Schrödinger's equation for a general quantum system, $i\partial_t \psi_t = \mathcal{H}\psi_t$. Here $\psi$ denotes the quantum state vector and $\mathcal{H}$ denotes the Hamiltonian operator of the system. We considered the following two types of Hamiltonians,

- For quantum oscillator, the Hamiltonian is given by $\mathcal{H} = \Delta + V$, where $\Delta$ is the Laplacian operator, and $V(x) = \frac{1}{2}\omega^2\|x - x_0\|^2$ is a random potential function, where $\omega$ and $x_0$ are random variables. Note that these are infinite dimensional objects. To obtain a valid element in $\mathsf{U}(n)$ for a finite $n$, we perform spectral discretization on the Laplacian operator as well as the random potential to get a finite-dimensional Hamiltonian operator $\mathcal{H}_h$, with which the time-evolution operator is computed with. We choose $t = 1$ in this case.

- For Transverse field Ising Model, the Hamiltonian is given by

$$\mathcal{H} = -\sum_{\langle i,j \rangle} J_{ij}\sigma_i^z\sigma_j^z - \sum_i g_i\sigma_i^x$$

  where $\sigma_i^z, \sigma_i^x$ are the Pauli matrices, $J_{ij}$ is the coupling parameter and $g_i$ is the field strength. Here $J_{ij}$ and $g_i$ are random variables, which corresponds to the situation of RTFIM. The time-evolution operator is generated with such a Hamiltonian at $t = 1$.

# H Additional Numerical Results

In this section, we present additional numerical results on learning time-evolution operators for ensembles of quantum systems, which are data distributions on complex-valued Unitary groups. We randomly select entries to scatter plot the data generated by TDM and compare it against the ground truth distribution. Additionally, we plot against the data distribution generated by RFM to further demonstrate our approach's advantage. As shown in Figure 10 and Figure 11, TDM captures the complicated patterns of distribution more accurately compared with RFM. While in general RFM also well captured the global shape of the data distribution, the details near the distribution boundary tend to be noisier than the one produced by TDM. This potentially originates from the approximations adopted by RFM, suggesting again the advantage of our proposed approximation-free TDM.

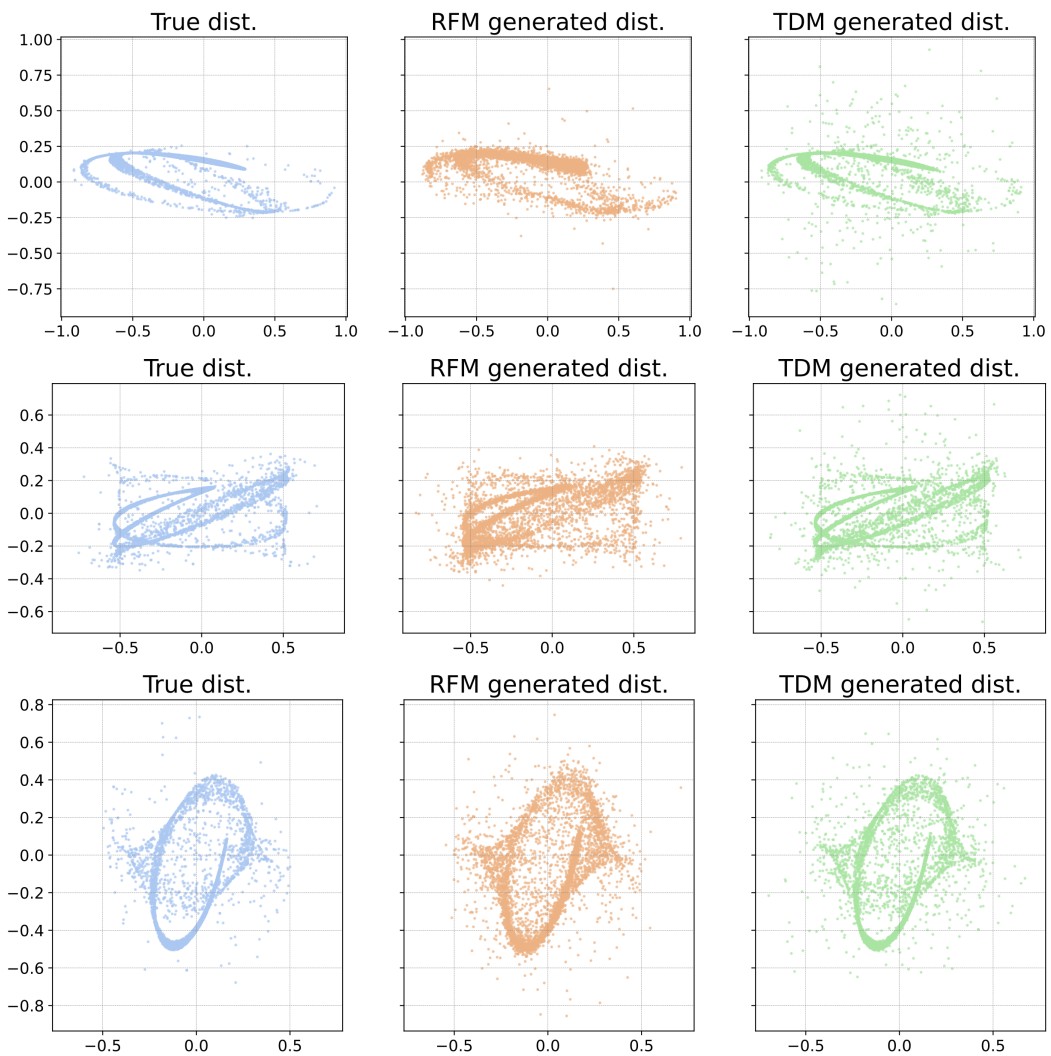

Figure 10: Visualization of Generated Time-evolution Operator of Quantum Oscillator on $\mathsf{U}(n)$ by TDM, compared against true distribution and RFM. Plotted entries are randomly chosen. **Top row**: $\mathsf{U}(4)$. **Mid row**: $\mathsf{U}(6)$. **Bottom row**: $\mathsf{U}(8)$

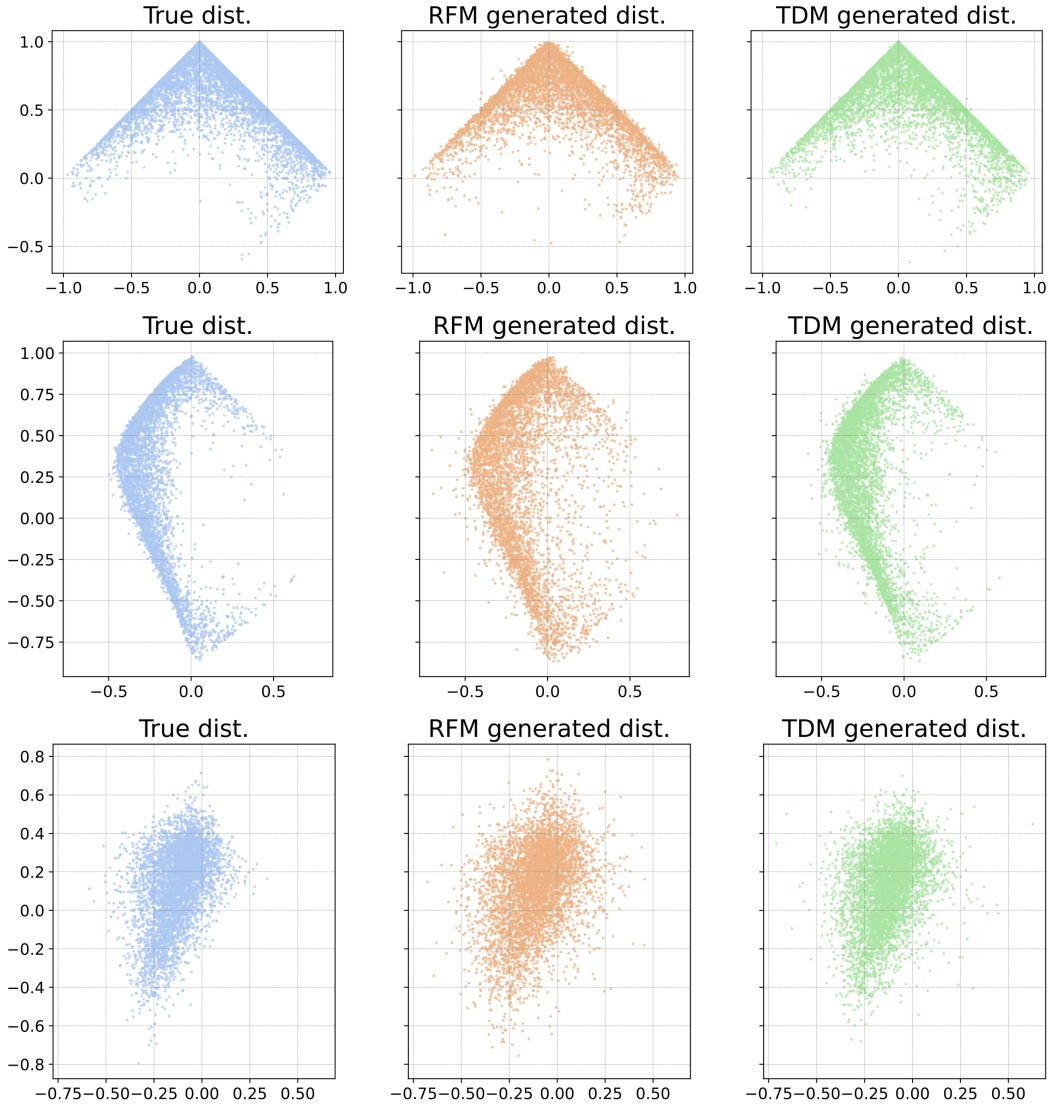

Figure 11: Visualization of Generated Time-evolution Operator of Transverse Field Ising Model (TFIM) on $\mathsf{U}(n)$ by TDM, compared against true distribution and RFM. Plotted entries are randomly chosen. **Top row**: $\mathsf{U}(4)$. **Mid row**: $\mathsf{U}(8)$. **Bottom row**: $\mathsf{U}(8)$

