# OpenReview forum: "Trivialized Momentum Facilitates Diffusion Generative Modeling on Lie Groups"
_ICLR.cc/2025/Conference — ICLR 2025 Poster_

### Official Review · Reviewer_dYfk · 2024-10-31

**Soundness:** 4
**Presentation:** 3
**Contribution:** 4
**Rating:** 8
**Confidence:** 4

**Summary:**

- The paper is concerned with tackling the challenge of incorporating diffusion models into a manifold setting
- Their key takeaway is that on Lie groups for both training and sampling diffusion can take place in the Lie algebra
- They show that their method performs well on various Lie groups of various dimensions that arise in different applications

**Strengths:**

- The authors provide a very nice motivation for the problem and highlight the key challenges that current manifold-valued models are dealing with.
- The mathematics is displayed in a very clear way and adequately addresses the challenge discussed in the motivation, i.e., how to set up manifold diffusion models on the Lie algebra. They also address different types of Lie groups well. In particular, it is nice that they highlight the special case of Abelian Lie groups learning, where training becomes easier than on non-Abelian ones
- The numerical experiments nicely display that the method performs well on high-dimensional Lie groups (which is where one would expect issues due to numerical inaccuracies from projections and manifold mappings), but also showcase that for low-dimensional Lie groups performance is superior to baselines.

**Weaknesses:**

- I feel that for the introduction, there are some mistakes in terms of English grammar
- Some parts could have been explained in more detail, but given the page limit restrictions it is natural that the authors aimed to be as concise as possible

**Questions:**

Questions for clarification
- p.2 l.59: the authors bring up numerical integration errors. It would be helpful to the reader to specify this a bit more. I assume this is because mappings like the exponential mapping can be expensive. Is this only the case in high dimensions or even on low-dimensional Lie groups?
    - a bit more nuance would improve the paper a lot as it would make the actual challenge even clearer, which would highlight your contributions even more
- p.2 l.78: the authors write ``while still encapsulating the curved geometry without any approximation.’’ There is an approximation error if there is curvature right?
    - It would be great if the authors are a bit more nuanced here as well as it should be clear that this method might introduce new errors.
- Section 3: In the Lie algebra, the learned distribution wraps around. So ideally, the score should be periodic or should become zero if this cannot be enforced. Is there some mechanism that enforces this?
    - If not, I would recommend adding this explicitly to let the reader know that there is possibility for error here.
- p.6 l.337: In the implicit score matching paragraph, it is not very clear to me how one would go about sampling here (does one have to do a forward simulation for this?). Would the authors clarify this?
- Section 5:
    - It feels natural to also mention something about flow matching in the outlook. Do the authors have any ideas on this they would like to disclose in the conclusions?

Additional feedback
- Section 1: please do a grammar check. A couple of small things that I noticed:
    - p.1 l.43: change ''manifold’’ in … Neural ODE to manifold with maximum … to ''manifolds’’. This reads better
    - p.1 l.44: change ''algorithm’’ in Rozen; Ben-Hamu develop simulation free algorithm but… to ''algorithms for the same reason as above
    - p.1 l.94: add ''a'' right before ''trivialized’’ in …through learning trivialized score function…
- Section 2: It looks a bit odd that section 2 is just one line. Consider integrating this line in section 3 instead.
- Section 3:
    - p.3 l.133: replace ''sequal'' with ''following''. It sounds like this will be discussed in your next paper rather than further down in this one.
    - p.3 l.147: the authors consider the tangent map. I believe that this is the differential? For the reader who is used to other nomenclature and/or notation, I would suggest to also mention that this is often referred to as the differential.
    - In equation 8, shouldn’t there be a superscript i for the xi in the dt term?
    - p.7 l.373: the authors write ``do not admit a closed form.’’ A closed form what?

---

> ### Author Response · Authors · 2024-11-20
> **Response to Reviewer dYfk**
>
> We thank the reviewer for the supportive review. We are glad the reviewer can appreciate our work, and highlight TDM as well-motivated and the paper as well-structured with clearly displayed mathematics and comprehensive numerical experiments. We provide the detailed response below to your comments and questions.
>
> **Weakness 1 (Grammar & Explanations)**
> > I feel that for the introduction, there are some mistakes in terms of English grammar
> > Some parts could have been explained in more detail, but given the page limit restrictions it is natural that the authors aimed to be as concise as possible
>
> We thank the reviewer for the efforts to improve the readability of this paper. We have improved the introduction section by doing grammar checks and also addressing the typos you have spotted in the revision. We also thank the reviewer for understanding that we couldn't explain everything in full detail due to page limit constraints. However, we are happy to help clarify any part of the paper here if you find that helpful.
>
> **Question 1 (Numerical Integration Errors)**
> > p.2 l.59: the authors bring up numerical integration errors. It would be helpful to the reader to specify this a bit more. I assume this is because mappings like the exponential mapping can be expensive. Is this only the case in high dimensions or even on low-dimensional Lie groups? a bit more nuance would improve the paper a lot as it would make the actual challenge even clearer, which would highlight your contributions even more
>
> Thanks for a great question. Without our trivialized momentum technique, the ODE to integrate is in general of the form $\dot{g} = f(g)$ for some $f$. Even if one uses the exponential map, namely $g_{k+1}=\exp_{g_k}(h f(g_k))$, it is still an approximation of the actual solution. On the other hand, our approach only needs the solution to $\dot{g}=g\xi$, where $\xi$ is a constant, and therefore the solution $g_{k+1}=g_k \text{expm}(h\xi)$ is not only staying on the manifold but also an exact update for the position variable $g$.
>
> **Question 2 (Approximation and Curvature)**
> > p.2 l.78: the authors write ``while still encapsulating the curved geometry without any approximation.’’ There is an approximation error if there is curvature right? It would be great if the authors are a bit more nuanced here as well as it should be clear that this method might introduce new errors.
>
> Thanks for the question. In our framework, there is no approximation error even though the Lie group is curved. This is because the curvature is already taken care of by the exponential map, which can be computed exactly by matrix exponential in our situation. We will make sure this is better clarified.
>
>
> **Question 3 (Wrapped Distribution)**
> > Section 3: In the Lie algebra, the learned distribution wraps around. So ideally, the score should be periodic or should become zero if this cannot be enforced. Is there some mechanism that enforces this? If not, I would recommend adding this explicitly to let the reader know that there is possibility for error here.
>
> In our framework, the distribution does not wrap around on Lie algebra since the Lie algebra is essentially an Euclidean space and the $\xi$ marginal of the probability distribution is simply Gaussian for any time $t$. The score function associated with the Lie algebra element $\xi$ is not periodic, and we do not need to enforce it, and therefore there is no extra source of approximation errors.
>
> **Question 4 (Sampling in ISM)**
> > p.6 l.337: In the implicit score matching paragraph, it is not very clear to me how one would go about sampling here (does one have to do a forward simulation for this?). Would the authors clarify this?
>
> Thanks for another great question. To evaluate the implicit score matching objective, one indeed needs to simulate the forward dynamics to generate samples approximately distributed as $p_t$. In practice, this is performed with Forward Operator-Splitting Integrator, which is explained in Algorithm 2 in Appendix D.1.
>
> **Question 5 (Relation to Flow Matching)**
> > Section 5: It feels natural to also mention something about flow matching in the outlook. Do the authors have any ideas on this they would like to disclose in the conclusions?
>
> Thanks for the good suggestion. We re-emphasized the comparison with RFM in the conclusion section of the revised version. As we have discussed and compared with Riemannian Flow Matching in both the literature review and experiments section, this makes the paper more coherent.
>
> **Other Questions (Typos)** We have fixed grammar errors and typos and revised the paper according to your suggestions to improve paper readability and clarity. Thank you for spending the effort to help us improve the paper!
>
> ----
> Hoping we clarified the misunderstandings and addressed the concerns, we thank you again for helping us improve the clarity of our writing. Please do kindly let us know if anything is still not clear.

---

> > ### Comment · Reviewer_dYfk · 2024-11-26
> > **Response to author**
> >
> > Many thanks to the authors for the detailed responses. Based on the replies and the updates in the revised manuscript, I feel that my initial grade is still a good reflection of the paper.

---

> > > ### Author Response · Authors · 2024-11-28
> > >
> > > Dear Reviewer dYfk,
> > >
> > > Thank you for your comments and support of our work! We deeply appreciate your efforts in improving our manuscript.
> > >
> > > Best,
> > >
> > > Authors

---

### Official Review · Reviewer_N41G · 2024-11-03

**Soundness:** 2
**Presentation:** 3
**Contribution:** 2
**Rating:** 6
**Confidence:** 3

**Summary:**

This paper provides a new framework for performing generative modelling on Lie groups which doesn’t require any geometry-specific projections or approximations, utilizing the notion of “trivialized momentum.” The authors provide the denoising score matching loss which can be used for SO(2)^n, as the conditional transition probabilities are tractable and an implicit score matching loss for other connected compact Lie groups. They also provide an integration scheme for the forward and backward dynamics.

**Strengths:**

- The paper is overall well-written and easy to follow. The problem the authors tackle is interesting and important.
- The proposed technique is interesting and avoids some of the problems mentioned with other existing approaches, such as projections, requiring specific architectures, etc.
- The mathematical details are explained cohesively and clearly.
- The results on the torus datasets, as well as the toy tasks for the SO(n) and U(n) groups seem promising.

**Weaknesses:**

- The main weakness of the paper is the lack of rigorous experiments to demonstrate the effectiveness of the model in comparison with existing approaches.
- First, the authors didn’t clarify how the experimental setup and training procedures vary across the different benchmarks. The experimental results would be more convincing if details on the setup are clearly provided.
- Secondly, it wasn’t clear me to what approximations one has to make when doing RFM in the torus settings, especially since going from the Lie algebra to the Lie group element is essentially trivial in that case. Could the authors clarify what approximations are required in RFM for SO(2)^n experiments?
- Lastly, although the results on the SO(n) and U(n) experiments are interesting and show promise, more rigorous experiments are required to show the scalability of the implicit score-matching loss (as it generally doesn’t perform as well). I think the paper could be strengthened if the authors clearly show better performance compared to RFM and RDM on the higher-dimensional SO(n) and U(n) tasks.
- It would be very helpful if the authors provided a more detailed discussion on the approximations made in RFM and RDM which the authors attribute to the worse performance of these models in the torus settings.
- The paper also provides experiments for the SO(3) setting, however it doesn’t provide a comparison with SE(3) generative models that rely on IGSO(3) distribution. As mentioned in the paper, the special structure of SO(3) can be leveraged, which makes applying TDM in that setting less motivated.

**Questions:**

See above.

---

> ### Author Response · Authors · 2024-11-20
> **Response to Reviewer N41G (Part I)**
>
> We thank the reviewer for the comments. We are glad the reviewer can recognize our contribution, highlighting our work as clearly structured, grounded in well-explained, solid mathematics, and our approach as novel and interesting. We provide a detailed response below to your comments. If further clarifications are needed, please feel free to engage in the discussion.
>
> **Weakness 1 (Lack of Experiment details):**
> > The main weakness of the paper is the lack of rigorous experiments to demonstrate the effectiveness of the model in comparison with existing approaches.
> > First, the authors didn’t clarify how the experimental setup and training procedures vary across the different benchmarks. The experimental results would be more convincing if details on the setup were clearly provided.
> > Although the results on the SO(n) and U(n) experiments are interesting and show promise, more rigorous experiments are required to show the scalability of the implicit score-matching loss (as it generally doesn’t perform as well). I think the paper could be strengthened if the authors clearly show better performance compared to RFM and RDM on the higher-dimensional SO(n) and U(n) tasks.
>
> We thank the reviewer for bringing this point up. To provide more details on the training procedures and make the numerical results more convincing, we expand the paragraph **"Architectural Framework"** and **"Training Hyperparameters"** in Appendix G.1 and include a detailed training configuration in the revision. We include the choice of all the important training hyperparameters and a more detailed explanation of the neural architectures employed in our experiments. We will also soon make the code implementation public, which contains full hyperparameter configurations for different experiments.
>
> To better present the advantage of our approach compared with existing approaches, we also include additional numerical results in Appendix H in the revision. We additionary compare TDM with RFM on two types of distribution on Unitary groups ranging from $\mathsf{U}(4)$ to $\mathsf{U}(8)$, namely the time evolution operator of Quantum oscillator and Random Transverse Field Ising Models. As is evident from the plots, TDM captures the complicated patterns of distribution more accurately compared with RFM. While in general RFM also well captured the global shape of the data distribution, the details near the distribution boundary tend to be noisier than the ones produced by TDM. This potentially originates from the approximations adopted by RFM, suggesting again the advantage of our proposed method.

---

> > ### Author Response · Authors · 2024-11-20
> > **Response to Reviewer N41G (Part II)**
> >
> > **Weakness 2 (Approximation in $\mathsf{SO}(2)$):**
> > > Secondly, it wasn’t clear me to what approximations one has to make when doing RFM in the torus settings, especially since going from the Lie algebra to the Lie group element is essentially trivial in that case. Could the authors clarify what approximations are required in RFM for SO(2)^n experiments?
> > > It would be very helpful if the authors provided a more detailed discussion on the approximations made in RFM and RDM which the authors attribute to the worse performance of these models in the torus settings.
> >
> > We thank the reviewer for bringing this question up and allowing us to further clarify. When we attribute the worse performances of RFM/RSGM to approximations, we mean it under the general Lie group setting. Indeed for $\mathsf{SO}(2)$, depending on the parameterization of this manifold, some special treatment can be employed to mitigate errors induced by approximations due to the curved geometry, but this could potentially lead to other issues. In the following, we will discuss the pros and cons of different parameterizations, and explain why some sources of errors remain in RFM/RSGM.
> >
> > 1. Parameterizating $\mathsf{SO}(2)$ as real 2x2 matrices. In such case, RFM/RSGM learns a score/vector field with outputs in $\mathbb{R}^{2\times2}$ and not guaranteed to be in $T\mathsf{SO}(2)$. Projection onto the tangent space is needed during inference to obtain a valid velocity, which incurs some approximation errors. Moreover, RFM adopts an Euler sampler, which moves samples along the tangent direction and causes the trajectory to leave the manifold. This necessitates the projection onto the manifold $\mathsf{SO}(2)$ after each discretization steps, which produce additional errors.
> > 2. Parameterizating $\mathsf{SO}(2)$ as interval $[0, 2\pi]$, with periodic boundary condition. In such a case, indeed approximation errors originate from tangent space, and Euler integration as discussed in point 1 above no longer exists. However, this leads to new issues as neural networks cannot easily learn periodicity and the learned score/velocity tends to be inconsistent near the interval boundary. This issue prevents these approaches from effectively capturing complicated patterns when adopting such a manifold parameterization. Similar observations have also been made in [Lou et al. 2023] (See Fig 2, 3 in [Lou et al. 2023] for demonstration).
> >
> > **Weakness 3 (Comparison on $\mathsf{SO}(3)$):**
> > > The paper also provides experiments for the SO(3) setting, however it doesn’t provide a comparison with SE(3) generative models that rely on IGSO(3) distribution. As mentioned in the paper, the special structure of SO(3) can be leveraged, which makes applying TDM in that setting less motivated.
> >
> > We agree with the reviewer that $\mathsf{SO}(3)$ may not be the best scenario that demonstrates the full capability of TDM, because indeed there is a simulation-free training approach for $\mathsf{SO}(3)$ due to its special structure and based on IGSO(3) distribution. But what we compared to in this example (RSGM) is already trained with denoising score-matching loss based on IGSO(3), and TDM still shows competitive performance. In addition, we want to comment that we included this example ($\mathsf{SO}(3)$) because we'd like to provide fair, comprehensive numerical experiments. We also included other examples such as $\mathsf{U}(n)$ where there is no denoising score matching-based diffusion model available.
> >
> > ----
> >
> > Hoping we clarified the misunderstandings and addressed the concerns, we thank you again for helping us improve the clarity of our writing. Please do kindly let us know if anything is still not clear.
> >
> > ----
> > References:
> >
> > Lou, Aaron, et al. Scaling Riemannian diffusion models. (2023)

---

> > > ### Comment · Reviewer_N41G · 2024-11-25
> > >
> > > Thank you for your response.
> > > Regarding the approximation errors in SO(2), even though I’m not sure how much of an issue it would be in practice for the network to learn the periodicity, I believe these arguments should be included in the paper to make it clearer. I am willing to increase my score from a 5 to a 6. However, I still believe that more rigorous experiments (and not just visualizations of toy tasks would be beneficial to make the results more convincing and showcase the benefits of the proposed method.

---

> > > > ### Author Response · Authors · 2024-11-26
> > > >
> > > > Dear Reviewer N41G,
> > > >
> > > > We deeply appreciate your response and your willingness to consider raising your score! Your specific suggestions have been instrumental in improving our manuscript.
> > > >
> > > > Thank you again for your valuable review and support of our work.
> > > >
> > > > Best regards,
> > > > Authors

---

### Official Review · Reviewer_AGbj · 2024-11-03

**Soundness:** 4
**Presentation:** 3
**Contribution:** 3
**Rating:** 8
**Confidence:** 3

**Summary:**

The paper introduces a new framework for constructing diffusion models on manifold-structured data in the particular case where the manifold is a Lie group. This is achieved through the use of a forward noising process defined by kinetic Langevin dynamics on the Lie group and its tangent space jointly. The authors demonstrate that the group structure of this class of manifolds allows for the corresponding reverse process to rely solely on a Euclidean score function, as well as allowing for the manifold-preserving and projection-free sampling of the forward and reverse processes, circumventing the complexities and multiple approximations required in previous work. The authors then apply this to various tasks presenting convincing results.

**Strengths:**

The authors present a novel and clever way to address the computational limitations of previous approaches to manifold-valued diffusion models, by exploiting the group structure of Lie groups through the use of Kinetic Langevin dynamics. This allows for learning a score function in a fixed Euclidean space which reduces the numerical errors associated with the approximations arising from scores defined directly on curved manifolds. This approach is well-fleshed out by the authors who provide a detailed methodology for the application of this for generative modelling, for example, by providing an integration scheme which is manifold-preserving and projection-free, and training objectives for both Abelian Lie groups and non-Abelian Lie groups. This framework is further validated by experiments demonstrating the novel scaling of manifold-valued diffusion models to SO(N), N>3.

**Weaknesses:**

The framework can only provide simulation-free training for the manifold SO(2) and its product space which is less flexible than Riemannian Flow Matching. However, this setting is still important for applications.

**Questions:**

Some possible typos in the text:
- line 155, missing brackets for g_t, \xi_t?
- line 325, should be "\sigma_t evaluated at"?
- line 1022, the A_\xi^\F should be A_\xi^\B?

---

> ### Author Response · Authors · 2024-11-20
> **Response to Reviewer AGbj**
>
> We sincerely thank the reviewer for the supportive review, recognizing our work as novel, convincing, and well-validated with comprehensive numerical experiments. We provide a detailed response to your comments below.
>
> **Weakness 1 (Training flexibility)**
> > The framework can only provide simulation-free training for the manifold SO(2) and its product space which is less flexible than Riemannian Flow Matching. However, this setting is still important for applications.
>
> Thank you very much for acknowledging that $\mathsf{SO}(2)$ and its product space are still important for applications. Regarding Riemannian Flow Matching, we would like to additionally comment that while it potentially enjoys simulation-free training in many cases, it comes with the cost of the need to compute the logarithm map for defining the conditional flow. Unlike the exponential map, the logarithm map as its inverse is much more numerically unstable and harder to compute. TDM does not require the computation of the logarithm map during training, so it is better in this regard, but it is indeed not always simulation-free. Therefore, which one is theoretically better is not yet completely clear to us.
>
> **Questions (Typos)**
>
> We have fixed grammar errors and typos and revised the paper according to your suggestions to improve paper readability and clarity. Thank you for spending this effort in improving the paper!

---

> > ### Comment · Reviewer_AGbj · 2024-11-26
> > **Response to rebuttal**
> >
> > Thanks to the authors for the response. I will maintain my score.

---

> > > ### Author Response · Authors · 2024-11-28
> > >
> > > Dear Reviewer AGbj,
> > >
> > > Thank you for your comments and support of our work! We deeply appreciate your efforts in improving our manuscript.
> > >
> > > Best,
> > >
> > > Authors

---

### Official Review · Reviewer_25XD · 2024-11-04

**Soundness:** 3
**Presentation:** 3
**Contribution:** 2
**Rating:** 5
**Confidence:** 4

**Summary:**

The authors present a method for diffusion on Lie groups. The method is based on the recently published idea of trivialized momentum, that is, diffusion (i.e. noising) is performed on the Lie algebra of the group, and not on the Lie group itself. The advantage of this approach is that the Lie algebra is a vector space isomorphic to $\mathbb{R}^n$, and therefore it simplifies the problem since all the curvature terms pretty much drop out. This diffusion is brought back into the actual Lie Group G by using the map connecting two tangent spaces of G at different point, which is the tangent of the left multiplication map.

The authors show that this noising dynamics has an inverse dynamics as well, in very familiar fashion as the usual diffusion dynamics, in which a score function appear, which it is possible to learn with neural networks.

The authors further argue that it is possible to have a dynamics that is exaclty on the curved manifold at all steps without the need of a projection (I do not agree with this statement, please refer below in the "Questions" section). This is done by separating each step into two contributions, i.e., the update of the Lie algebra element $d\xi$ and the update of the lie group element $dg = g \xi$. Note that also this procedure is already known in the literature.

They provide an explicit formula for the Abelian case, while they point out that for the general case, it is necessary to learn the score with implicit score matching, which unfortunately involves the divergence term.

**Strengths:**

The topic is very relevant, as diffusion-based models achieved quite successful results in many applications, but very little has been explored in terms how inductive bias can be incorporate into them, and diffusion on manifold is surely an important field.

The notation is fairly consistent and the math is, to the extend that I checked, correct (I did not carefully checked the proofs in the appendix).

**Weaknesses:**

My two main concerns regard the originality of the work and its practical relevance:

- **Originality**: As far as I can see, the idea and the theory behind trivialized momentum has been developed in previous works, as well as the  splitting discretization technique to exactly integrate the SDEs. The authors prove a theorem for the inverse SDE of the system, but given that the non-trivial equation (about the variation of the Lie algebra) is essentially in flat space, it reduces pretty much to the flat-space formula.
- **Practical Relevance**: The main drawback is in the non-Abelian case, which is the most interesting since the Abelian case is essentially SO(2) (or copies of it). For non-Abelian groups, there's no conditional transition probability available, requiring implicit score matching. This involves computing the divergence with respect to the data space (more precisely, the Lie algebra space, but as dim data = dim $G$ = dim $\mathfrak{g}$ its computationally as expensive). This limitation significantly reduces the method's practicality and makes it unlikely to be widely adopted.

I think the paper would be significantly more relevant if it provided an analogue of Theorem 2 for non-Abelian groups.

**Questions:**

- 280-283: The sentence “Since the Brownian motion…” does not seem to be grammatically correct.
- 292: “Unfortunately, note [that] even though”. Missing “that”.
- 382: $A_g^B$ used twice, probably copy-paste error.
- The authors make several times the claim that the formalism on which they based their work does not required projection on the manifold. I think this statement, as is, is false. Group equation is $dg = g_0 \xi$, which is an equation on the tangent space of the group. Its solution is (neglecting the variation of $\xi$), $g=\exp(g_0 \xi)$. But the exponential map is precisely the projection function on Lie groups, $\exp: TG —> G$. So to stay on the manifold the authors are performing a projection exactly as all other methods too.

---

> ### Author Response · Authors · 2024-11-20
> **Response to Reviewer 25XD (Part I)**
>
> We thank the reviewer for the comments. We are glad the reviewer can recognize our contribution to manifold diffusion models. We provide the detailed response below. If further clarifications are needed, please feel free to engage in the discussion.
>
> **Weakness 1 (Originality):**
> > As far as I can see, the idea and the theory behind trivialized momentum have been developed in previous works, as well as the splitting discretization technique to exactly integrate the SDEs.
>
> Previous works employed trivialized momentum for optimization and sampling, and our work is the first that uses it in generative modeling. We're unsure if the novelty of generative modeling algorithm should be criticized for gaining inspiration from existing sampling/stochastic analysis literature. For example, score-based generative model (SGM) [e.g., Yang et al. 2020] turned Ornstein–Uhlenbeck process into a generative model, Critically-damped Langevin Diffusion (CLD) [Dockhorn et al. 2021] turned kinetic langevin dynamics into a generative model. We think they had significant originality and impact, and our case is the same: TDM turned trivialized Lie-group sampling dynamics into a generative model.
>
> Regarding splitting discretization, similarly, while this technique has been studied in the literature of numerical analysis, we are the first to use it for Lie group generative modeling. We also do not claim any credit over its convergence analysis, and present it in the paper just for self-consistency.
>
>
> > The authors prove a theorem for the inverse SDE of the system, but given that the non-trivial equation (about the variation of the Lie algebra) is essentially in flat space, it reduces pretty much to the flat-space formula.
>
> Regarding Theorem 1 in the paper (which is about the reverse process of TDM), we agree that the result itself is not surprising. That is why our approach is good -- it just gives results like in the Euclidean case, which is the intention of our design. However, the proof of Theorem 1 (see Appendix B) is not straightforward due to the curved geometry. In addition, we also presented many other novel and non-trivial theoretical results such as Theorem 2 and Theorem 5.
>
> **Weakness 2 (Practical Relevance):**
> > The main drawback is in the non-Abelian case, which is the most interesting since the Abelian case is essentially SO(2) (or copies of it). For non-Abelian groups, there's no conditional transition probability available, requiring implicit score matching. This involves computing the divergence with respect to the data space (more precisely, the Lie algebra space, but as dim data = dim $G$ = dim $\mathfrak{g}$ its computationally as expensive). This limitation significantly reduces the method's practicality and makes it unlikely to be widely adopted.
>
> We thank the reviewer for pointing it out. We agree with the reviewer that the extension of denoising score matching (DSM) to general non-Abelian Lie groups would be exciting. However, we also note that existing Riemannian Diffusion Models, such as RSGM [De Bortoli et al. 2022] and RDM [Huang et al. 2022], also heavily rely on implicit score matching when dealing with general Lie groups. In fact, the success on simulation-free training of manifold diffusion models has been very limited in the literature, and we're unaware of any denoising score-matching result for **general** non-Abelian groups.

---

> > ### Author Response · Authors · 2024-11-20
> > **Response to Reviewer 25XD (Part II)**
> >
> > **Question 4 (Projection):**
> > > The authors make several times the claim that the formalism on which they based their work does not require projection on the manifold. I think this statement, as is, is false. The group equation is $dg = g_0 \xi$, which is an equation on the tangent space of the group. Its solution is (neglecting the variation of $\xi$), $g = \exp(g_0 \xi)$ But the exponential map is precisely the projection function on Lie groups, $\exp: TG \rightarrow G$. So to stay on the manifold the authors are performing a projection exactly as all other methods too.
> >
> > We thank the reviewer for bringing it up and allowing us to further clarify. **Projection onto the manifold** in our setting means an operator $P: \mathbb{R}^{n} \rightarrow G$ that projects a point outside the Lie group $G$ to the Lie group. This notion is deployed in the literature to address the numerical errors led by inexact numerical integration of dynamics that cause the trajectory to leave the manifold. For example, given dynamics $\dot{g}=g\xi$ with constant $\xi$ in the Lie algebra, standard Euler scheme $g \mapsto g+h g \xi$ will no longer be on the manifold, and it is common to use $\hat{g_h}=P(g_0+h g_0\xi)$, i.e. an extra projection operation to put it back on the manifold. Here $\hat{g_h}$ would be an approximation of the exact solution $g_h=g_0 \text{expm}(h\xi)$, and the way it is obtained is an example of a retraction operation enabled by a projection. Projection by itself however does not simulate the dynamics. On the contrary, the exponential map $\exp_g: T_gG \rightarrow G$ is the operator that generalizes the notion of "moving in a straight line" to Lie groups, thus is different from the projection operator. In the case of $\dot{g}=g\xi$ with constant $\xi$, $g_h=g_0 \text{expm}(h\xi)=\exp_{g_0}(hg_0\xi)$ is the exact solution, where $\exp$ is the exponential map and $\text{expm}$ is standard matrix exponential that implements the exponential map, and no projection is involved.
> >
> > **Other questions (Typos):** We have fixed grammar errors and typos you spotted and updated in the revision. Thank you for spending this effort in improving the paper!
> >
> > ----
> >
> > Hoping we clarified the misunderstandings and addressed the concerns, we thank you again for helping us improve the clarity of our writing. Please do kindly let us know if anything is still not clear.
> >
> > ----
> > **Reference:**
> >
> > Song, Yang, et al. Score-based generative modeling through stochastic differential equations (2020)
> >
> > Dockhorn, Tim, Arash Vahdat, and Karsten Kreis. Score-based generative modeling with critically-damped langevin diffusion (2021)
> >
> > De Bortoli, Valentin, et al. Riemannian score-based generative modelling (2022)
> >
> > Huang, Chin-Wei, et al. Riemannian diffusion models (2022)

---

> > > ### Comment · Reviewer_25XD · 2024-11-26
> > >
> > > I thank the authors for their response.
> > >
> > > In Riemannian geometry a projection map is a map $P$ between $P:TM\rightarrow M$, where $M$ is a Riemannian manifold. In case the Riemannian manifold is a Lie group we have $P:TG\rightarrow G$, where $TG=\mathfrak{g}$ is the Lie algebra. The exponential map is exactly one of such maps. I did not claim that there do not exist other projection maps possible, but merely that the exponential map is one of them.
> > >
> > > If the authors want to make the case that other curved space methods necessitate a different kind of projection, they should make the specific case of why their projection is more advantageous, but the statement that the method does not need a projection is not correct.

---

> > > > ### Author Response · Authors · 2024-11-28
> > > >
> > > > Dear Reviewer 25XD,
> > > >
> > > > Thanks very much for your response!
> > > >
> > > > > In Riemannian geometry a projection map is a map $P$ between $P: TM \rightarrow M$, where $M$ is a Riemannian manifold.
> > > >
> > > > We feel like the reviewer's confusion may have arisen from the fact that there are **multiple notions of projections** and we have not been on the same page regarding it throughout the discussion. Therefore, please allow us to clarify these notions below:
> > > >
> > > > 1. One of them, sometimes called "natural projection", is indeed a common operation in differential geometry (note: this operation is not specialized to Riemannian geometry as no Riemannian structure is needed). This operation is the following mapping
> > > > $\pi: TM\to M$, $(x, v)\mapsto x$ for $x\in M$ and $v\in T_x M$
> > > > (see e.g., [1] page 382).
> > > > Here $TM$ is the tangent bundle, defined as $TM = \underset{x \in M}{\cup} \{x\} \times T_{x} M$ and $T_{x}M$ is the tangent space of M at $x$.
> > > > This operation bears no computational cost and is not what we have been discussing throughout the manuscript.
> > > >
> > > > 2. Another version is what we have been referring to: when $M$ is embedded in $\mathbb{R}^n$, the mapping
> > > > $P: \mathbb{R}^n\to M$: $P(x)=\arg\min_{y\in M} ||x-y||$
> > > > is also known as a projection. It is standard and widely adopted in the literature of, e.g., manifold optimization, sampling theory, and generative modeling, and therefore we did not expand on details in the manuscript. In fact, the defining formula above is quoted from a celebrated manifold generative modeling work ([2], Eq. 27; more references include other milestones in manifold optimization [3] and sampling [4]). Just in case more mathematical rigor is needed, a more formal definition is: given an embedding $i$ of $M$ in $\mathbb{R}^n$, the map $P: \mathcal{B}(i(M)) \to M$ from a neighborhood of the manifold to itself, defined as $P(x)=\arg\min_{y\in M} || x-i(y) ||$, is a projection.
> > > > This version of projection, however, often bears a computational cost, and it is what we try to reduce.
> > > >
> > > > [1] JM Lee. Introduction to Riemannian Manifolds. Springer 2018
> > > >
> > > > [2] RT Chen and Y Lipman. Flow matching on general geometries. ICLR 2024.
> > > >
> > > > [3] H Zhang, SJ Reddi, S Sra. Riemannian SVRG: Fast Stochastic Optimization on Riemannian Manifolds. NIPS 2016
> > > >
> > > > [4] S Bubeck, R Eldan, and J Lehec. Sampling from a log-concave distribution with projected Langevin Monte Carlo. Discrete & Computational Geometry, 2018
> > > >
> > > > > The exponential map is exactly one of such maps. I did not claim that there do not exist other projection maps possible, but merely that the exponential map is one of them.
> > > >
> > > > Assuming that the reviewer was thinking that the exponential map is the projection of the first kind due to his notation used, may we point out that **while natural projection is a $TM \to M$ mapping, not all $TM \to M$ mappings are natural projections**? In order to be a valid projection of the first kind, the operation must preserve the based point $x$ (see again, e.g., [1] page 382). While the exponential map is also a $TM \to M$ mapping, it does not preserve this base point and thus does not suit this definition of projection.
> > > >
> > > > If there is still a belief that the exponential map is a projection, maybe there is a third type of projection that we are unaware of. In this case, we would deeply appreciate some popular references.
> > > >
> > > > > In case the Riemannian manifold is a Lie group we have $P:TG\to G$, where $TG=\mathfrak{g}$ is the Lie algebra.
> > > >
> > > > $TG$ is the tangent bundle of $G$, which collects all tangent spaces along with their base points. $T_g G$ for any $g\in G$ is a tangent space at point $g$. $T_e G$ (not the tangent bundle $TG$) is the Lie algebra $\mathfrak{g}$, where $e$ is one special element of the group $G$, namely the identity element. These three are very different things, and our key contribution is to leverage the Lie algebra $T_e G$ instead of a standard treatment based on $TG$. We're not completely sure if this discussion is still relevant, but if there is something the reviewer would like us to do, we are happy to, but we'd appreciate it if the reviewer could further clarify the inconsistent notations so that we know our task.
> > > >
> > > > ---
> > > > In summary, we will be more than happy to clarify the different definitions of projections in our manuscript if that's what the reviewer is suggesting (thank you for pointing out). However, we are uncertain if this solely contributes to the reviewer's evaluation of our work as only worth a score of 3.
> > > >
> > > > The reviewer's initial review included two weaknesses, three questions on typos, and one question on projection, which we greatly appreciated and carefully explained/addressed in our first-round responses. Since the second round of comments was only about projection and we absolutely would like to make sure our paper and explanations make sense, may we ask if we managed to address the other concerns?
> > > >
> > > > Thank you again for engaging in the discussion and we are looking forward to your response!
> > > >
> > > > Best regards

---

> > > > > ### Comment · Reviewer_25XD · 2024-11-28
> > > > >
> > > > > I thank the authors for their detailed answer.
> > > > >
> > > > > For clarification, here is why I mean that the exponential map in this context can be considered a projection. Given a point $x\in G$, then we perform an update by shifting $x$ by an element of a Lie algebra $\xi \in \mathfrak{g}$. This does not make sense on $G$ since we cannot simply add $g + \xi$. We can however think of the embedding $G\subseteq \mathbb{R}^n$ (this can be done by thinking of $G$ as a manifold). Then, in the ambient space, we can formally perform the operation $g + \xi$ (by abuse of notation I use the same symbols even though I would need to define the lift to the ambient space coordinates). However, the new point $g+\xi \notin G$ anymore, so we need a projection to go back to $G$. Thus specifically this is a projection very similar to your second case, that is (thinking in terms of ambient space) $\exp : \mathbb{R}^n \rightarrow G, g + \xi \mapsto g' = \exp_g(\xi)$. The function to minimise is not the $L_2$ distance in the ambient space, but it nonetheless has a cost (unlike the natural projection).
> > > > >
> > > > > In summary, I would encourage the authors to clarify in the main text exactly what they need by projection (I understand that [1] makes the same claim but taking over claims from other sources is not always a good practice.)
> > > > >
> > > > > [1] Kong at at., "Convergence of Kinetic Langevin Monte Carlo on Lie groups"

---

> > ### Comment · Reviewer_25XD · 2024-11-28
> >
> > I thank the authors for their clarifications. My concerns about the novelty and the practical relevance still remain at large, but I acknowledge that the previous works did not have a focus on generative modeling. For this reason I will increase my score.

---

### Meta-Review · Area_Chair_gj5x · 2024-12-20

**Metareview:**

The paper studies diffusion processes on Lie groups by introducing the diffusion process on corresponding Lie algebra. This allows introducing the diffusion processes and training corresponding generative models on SO(n) and U(n) manifolds. All the reviewers have recognized the theoretical contribution of the paper. Given the high interest in generative modelling, the paper will be of high interest to the ICLR audience.

**Additional Comments On Reviewer Discussion:**

Several reviewers raised concerns regarding the proposed approach's scalability and practical evaluation. However, the merit of the proposed theoretical developments outweighed the issues of the empirical study.

---

### Decision · Program_Chairs · 2025-01-22

Accept (Poster)